# Mechanisms of resistance to VHL loss-induced genetic and pharmacological vulnerabilities

Jianfeng Ge[1,2], Shoko Hirosue [1], Leticia Castillon [3], Saroor A Patel [1,4], Ludovic Wesolowski[1], Anna Dyas[1,2], Cissy Yong[5,6], Sanne de Haan[7], Jarno Drost [7,8], Grant D Stewart [5], Anna C Obenauf [9], Daniel Muñoz-Espín [2,10 ✉] & Sakari Vanharanta [1,3,11✉]

## Abstract

The von Hippel-Lindau tumor suppressor (VHL) is a component of a ubiquitin ligase complex that controls cellular responses to hypoxia. Endogenous VHL is also utilized by proteolysis-targeting chimera (PROTAC) protein degraders, a promising class of anti-cancer agents. VHL is broadly essential for cell proliferation, yet it is a key tumor suppressor in renal cell carcinoma. To understand the functional consequences of *VHL* loss, and to identify targeted approaches for the elimination of *VHL* null cells, we have used genome-wide CRISPR-Cas9 screening in human renal epithelial cells. We find that, upon VHL loss, the HIF1A/ARNT complex is the central inhibitor of cellular fitness, suppressing mitochondrial respiration, and that VHL null cells show HIF1A-dependent molecular vulnerabilities that can be targeted pharmacologically. Combined VHL/HIF1A inactivation in breast and esophageal cancer cells can also provide resistance to ARV-771, a VHL-based bromodomain degrader that has anti-cancer activity. HIF1A stabilization can thus provide opportunities for early intervention in neoplastic *VHL* clones, and the VHL-HIF1A axis may be relevant for the development of resistance to the emerging class of PROTAC-based cancer therapies.

**Keywords** von Hippel-Lindau Tumor Suppressor (VHL); Renal Cancer; PROTAC; HIF1A; CRISPR/Cas9 Screening
**Subject Categories** Cancer; Urogenital System

## Introduction

Heterozygous germline mutations in the von Hippel-Lindau tumor suppressor (*VHL*) lead to a tumor predisposition syndrome that is characterized by the development of phaeochromocytomas, renal cell carcinomas, hemangioblastomas, and pancreatic neuroendocrine tumors (Kaelin, 2007). Biallelic inactivation of *VHL* is also the tumor-initiating genetic event in ~90% of sporadic clear cell renal cell carcinomas (ccRCCs) (Turajlic et al, 2018). The VHL protein functions as a substrate recognition subunit of an E3 ubiquitin ligase complex, the best-characterized substrates of which are the hypoxia-inducible factors HIF1A and HIF2A (HIFA). HIFA bind VHL when two of their conserved proline residues are hydroxylated (Kaelin, 2007). The HIFA prolyl hydroxylases EGLN1-3 require oxygen as a co-substrate, making them active only in normoxic conditions. Under hypoxia or in the absence of functional VHL, HIFA accumulates and dimerizes with HIF1B/ARNT, forming a helix-loop-helix transcription factor (Kaelin, 2007).

The endogenous ubiquitin ligase target recognition function of VHL can in *VHL* wild-type cancers, i.e., most cancers apart from ccRCC, be exploited by the VHL-dependent PROTACs, small molecules that can be engineered to degrade specific endogenous proteins through the E3 ubiquitin ligase pathway (Békés et al, 2022). These compounds serve as molecular bridges between a target protein of interest and an E3 ligase, such as the VHL complex, which directs them for proteasomal degradation. For example, ARV-771, a bromodomain degrader, and AU-15330, a SMARCA2/4 degrader, are dependent on endogenous VHL, and they have shown anti-tumor activity in experimental *VHL* wild-type cancer models (Xiao et al, 2022; Raina et al, 2016).

Even though the cellular consequences of VHL inactivation are relevant in several clinically independent contexts, they remain incompletely understood. For example, acute VHL loss is detrimental to cell proliferation (Young et al, 2008), yet *VHL* is a tumor suppressor in renal cancer (Young et al, 2008; Welford et al, 2010; Turajlic et al, 2018). As the development of renal cancer from *VHL* mutant renal epithelial cells can take decades (Mitchell et al, 2018), VHL loss-induced vulnerabilities could be exploited for early cancer intervention strategies, especially in high-risk individuals, such as *VHL* mutation carriers. Biallelic *VHL* inactivation in the germline can also lead to a severe systemic metabolic disorder (Perrotta et al, 2020). Such patients could benefit from therapies that limit the negative fitness effects caused by *VHL* inactivation.

[1]MRC Cancer Unit, University of Cambridge, Hutchison/MRC Research Centre, Cambridge, UK. [2]Early Cancer Institute, Department of Oncology, University of Cambridge, Cambridge, UK. [3]Translational Cancer Medicine Program, Faculty of Medicine, University of Helsinki, Helsinki, Finland. [4]Wellcome Sanger Institute, Cambridge, UK. [5]Department of Surgery, University of Cambridge, Cambridge Biomedical Campus, Cambridge, UK. [6]Cambridge University Hospitals NHS Foundation Trust, Cambridge, UK. [7]Princess Máxima Center for Pediatric Oncology, Oncode Institute, Heidelberglaan 25, Utrecht 3584 CS, The Netherlands. [8]Division of Cell Biology, Metabolism & Cancer, Department of Biomolecular Health Sciences, Faculty of Veterinary Medicine, Utrecht University, Utrecht, The Netherlands. [9]Research Institute of Molecular Pathology, Vienna Biocenter, Vienna, Austria. [10]Thoracic Cancer Programme, CRUK Cambridge Centre, Cambridge, UK. [11]Department of Biochemistry and Developmental Biology, Faculty of Medicine, University of Helsinki, Helsinki, Finland. ✉E-mail: dm742@cam.ac.uk; sakari.vanharanta@helsinki.fi

Finally, the role of endogenous VHL as a key player in PROTAC-induced anti-cancer effects (Békés et al, 2022) suggests that understanding the consequences of VHL loss could be important for understanding and reducing the likelihood of PROTAC-resistance in *VHL* wild-type cancers.

Using experimental cell line systems and large-scale CRISPR-Cas9 functional screening, we set out to investigate the mechanisms of VHL loss-induced proliferative suppression. We find that the HIF1A/ARNT complex is the central mediator of reduced fitness following VHL loss, and that VHL null cells display HIF1A-dependent genetic vulnerabilities that can be pharmacologically targeted. Moreover, we demonstrate that combined loss of VHL and HIF1A can provide a fitness advantage to human cancer cells under treatment with a VHL-dependent PROTAC. These results shed light on the functional consequences of VHL loss at the genome-wide scale and provide a proof of principle that VHL loss-dependent phenotypes can be pharmacologically targeted.

## Results

### HIF1A/ARNT-mediated proliferative suppression upon *VHL* inactivation

Given the important role of VHL in several clinically relevant contexts, we set out to study the functional consequences of VHL loss. First, we analyzed the large-scale Cancer Dependency Map CRISPR/Cas9 loss-of-function screening data set, which showed that apart from ccRCC cells, most of which already carry biallelic mutations in the *VHL* gene, cancer cell lines across different non-renal lineages were universally sensitive to VHL inactivation (Fig. 1A). This indicated that *VHL* mutant ccRCCs have escaped the negative fitness effect that accompany VHL inactivation in most cell lineages. Therefore, to investigate the effects of VHL loss in *VHL* wild-type human cells that have not evolved to resist the negative fitness effects of VHL loss, we used CRISPR-Cas9 to mutate *VHL* in the HK2 human renal epithelial cells, an immortalized cell line derived from human renal epithelial cells, and in human renal epithelial organoids. In line with the cancer cell line CRISPR/Cas9 screening data, *VHL* inactivation in HK2 cells led to strong inhibition of proliferation and an altered cellular morphology (Fig. EV1A–C). Reintroduction of sgRNA-insensitive *VHL* cDNA completely rescued the proliferation defect caused by CRISPR-Cas9-mediated *VHL* inactivation, confirming the specificity of the approach (Fig. EV1D,E). Unlike previously reported in mouse fibroblasts (Welford et al, 2010), reduced oxygen level did not rescue the effects of *VHL* loss (Fig. EV1F), even though it reduced the proliferative capacity of *VHL* wild-type cells in longer-term assays (Fig. EV1G). Inactivation of *VHL* also reduced the size of human renal epithelial organoids, but the cells still formed similar structures as wild-type cells (Figs. 1B and EV1H). We used a genetic GFP-dependent HIFA reporter to confirm HIF activity, a predicted downstream effect of VHL loss, in the VHL-engineered organoids, while wild-type organoids remained GFP negative (Fig. 1C).

To understand mechanisms of VHL loss-induced fitness loss, and to identify potential VHL loss-induced gene dependencies, we performed a genome-wide CRISPR-Cas9 loss-of-function screen in HK2 cells. To avoid the possibility of cells adapting to VHL inactivation during propagation before the screen started and to obtain enough VHL null cells despite their reduced proliferative

capacity, we engineered cell clones in which VHL expression could be regulated by doxycycline (dox), and endogenous *VHL* was either intact (WT8 control cells) or knocked out (MUT10 and MUT35) (Fig. EV2A). WT8 cells proliferated well regardless of dox, but MUT10 and MUT35 cells proliferated significantly better when dox, and consequently VHL, was present (Fig. EV2B–D). VHL expression was also associated with HIF1A and HIF2A degradation, as expected (Fig. EV2E). Cas9 editing efficiency was confirmed in all three clones (Fig. EV2F) before proceeding to the genome-wide screen (Fig. 1D). The sgRNA distribution of the two *VHL* mutant clones showed strong correlation at the endpoint (Fig. EV3A). When comparing sgRNA representation in VHL null cells at the end of the screen and pre-dox withdrawal, constructs targeting HIF1A, a known target of VHL, and its dimerization partner ARNT were clearly the most significantly enriched, with little evidence of consistent enrichment of constructs targeting other genes (Fig. 1E). Consistently, all five sgRNAs targeting HIF1A and ARNT showed enrichment (Fig. EV3B,C). *VHL* null cells were sensitive to the inhibition of several pathways, such as glycolysis, oxidative phosphorylation, MTORC1 signaling, DNA repair, Myc and E2F targets and the G2M checkpoint (Fig. EV3D), pathways that are typically important for proliferating cells, indicating that these cells, despite their reduced proliferative capacity in comparison to *VHL* wild-type cells, performed as expected in the pooled screen. Fluorescence-assisted cell sorting-based competition assays confirmed that HIF1A and ARNT inactivation, but not HIF2A inactivation, rescued the proliferation defect caused by *VHL* loss (Figs. 1F and EV3E–H). HIF1A and ARNT inactivation also restored the morphological appearance of *VHL* null cells (Fig. EV3I). Finally, HIF1A inactivation could rescue proliferation in *VHL* mutant human renal epithelial organoids (Fig. 1G,H), demonstrating the relevance of our observation in primary human cells. Overall, these results suggest that the HIF1A/ARNT complex is a central mediator of VHL loss-induced proliferative suppression.

### HIF1A-dependent mitochondrial inhibition in *VHL* null cells

To understand how gene expression was affected by *VHL* inactivation, we performed transcriptomic analysis by RNA-seq on MUT10, MUT35 and WT8 cells with and without dox withdrawal. As expected, WT8 cells showed little change in gene expression (Fig. EV4A; Dataset EV1). However, combined analysis of MUT10 and MUT35 cells revealed 220 genes significantly upregulated and 63 genes significantly downregulated upon dox withdrawal and consequent VHL depletion (Figs. 2A and EV4B–E; Datasets EV2 and 3). Gene set enrichment analysis of the transcriptomic data showed that genes in categories such as hypoxia, TNF-alpha signaling, glycolysis and epithelial-to-mesenchymal transition (EMT) were upregulated, and oxidative phosphorylation, MYC targets, G2M checkpoint and E2F targets were suppressed in VHL depleted cells (Fig. 2B). Although the expression of glycolysis genes is upregulated in *VHL* mutant cells, oxidative phosphorylation in mitochondria is the main source of ATP, suggesting that the proliferation defect of *VHL* mutant cells could be linked to suppressed oxidative phosphorylation (Fig. 2C). Oxygen consumption rate (OCR) in MUT10 and MUT35 cells in the absence of dox was reduced when compared to WT8 cells, with OCR value being ~90% lower than in wild-type cells (Fig. 2D,E). In

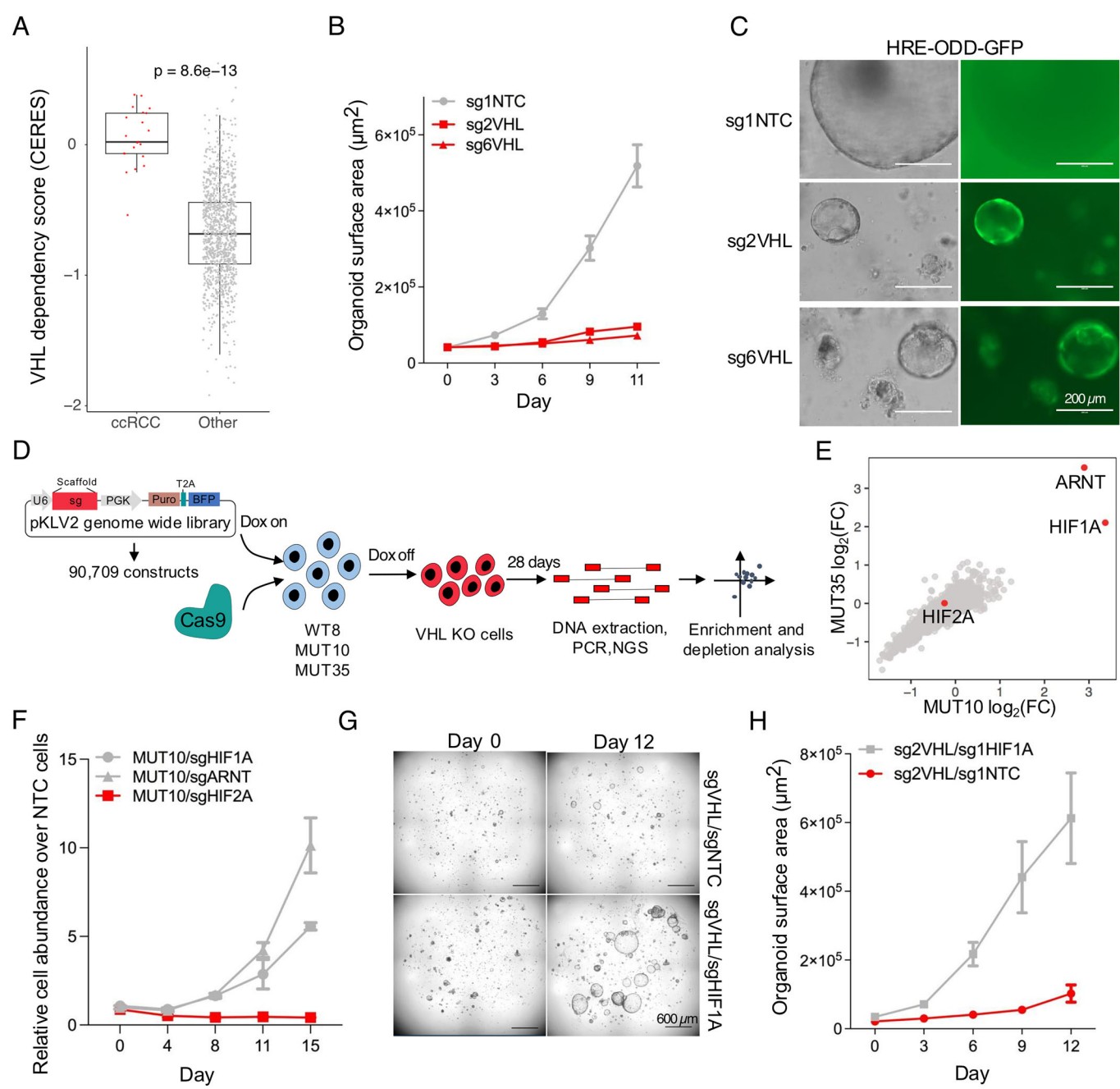

**Figure 1. Universal inhibition of cell proliferation upon *VHL* inactivation.**

(A) VHL dependency score in ccRCC cell lines ($N = 21$) and other pan cancer lines ($N = 1162$). Unpaired Wilcoxon test to calculate significance. The horizontal line in the box marks the median (Q2), the upper and lower hinges correspond to the 25th (Q1) and 75th (Q3) percentiles. The whiskers extend to 1.5× the interquartile range from Q1 to Q3. (B) Quantification of normal renal organoid growth (EMKC016) with and without VHL inactivation, $N = 32$ random growing organoids per condition and time point (mean and S.E.M.). (C) Activity of a hypoxia reporter (HRE-ODD-GFP) in normal renal organoids (EMKC016) with and without VHL deletion as determined by fluorescence microscopy. (D) Schematic of the pooled CRISPR-Cas9 screening strategy. (E) Gene level CRISPR-Cas9-based loss of function screening data. Beta scores showing change in sgRNA construct abundance in VHL mutant cells. $N = 2$ replicates per condition. (F) CRISPR-Cas9 based competition assay in VHL mutant MUT10 cells. HIF1A, HIF2A and ARNT mutant cells competed against cells transduced with non-targeting control constructs (NTC). Two sgRNAs per gene combined, $N = 3$ technical replicates per condition (mean and S.E.M.). (G) VHL mutant human renal epithelial organoids with or without HIF1A inactivation at different time points. (H) Quantification of organoid growth from (G) over time, $N = 13$ random growing organoids per condition and time point (mean and S.E.M.). Source data are available online for this figure.

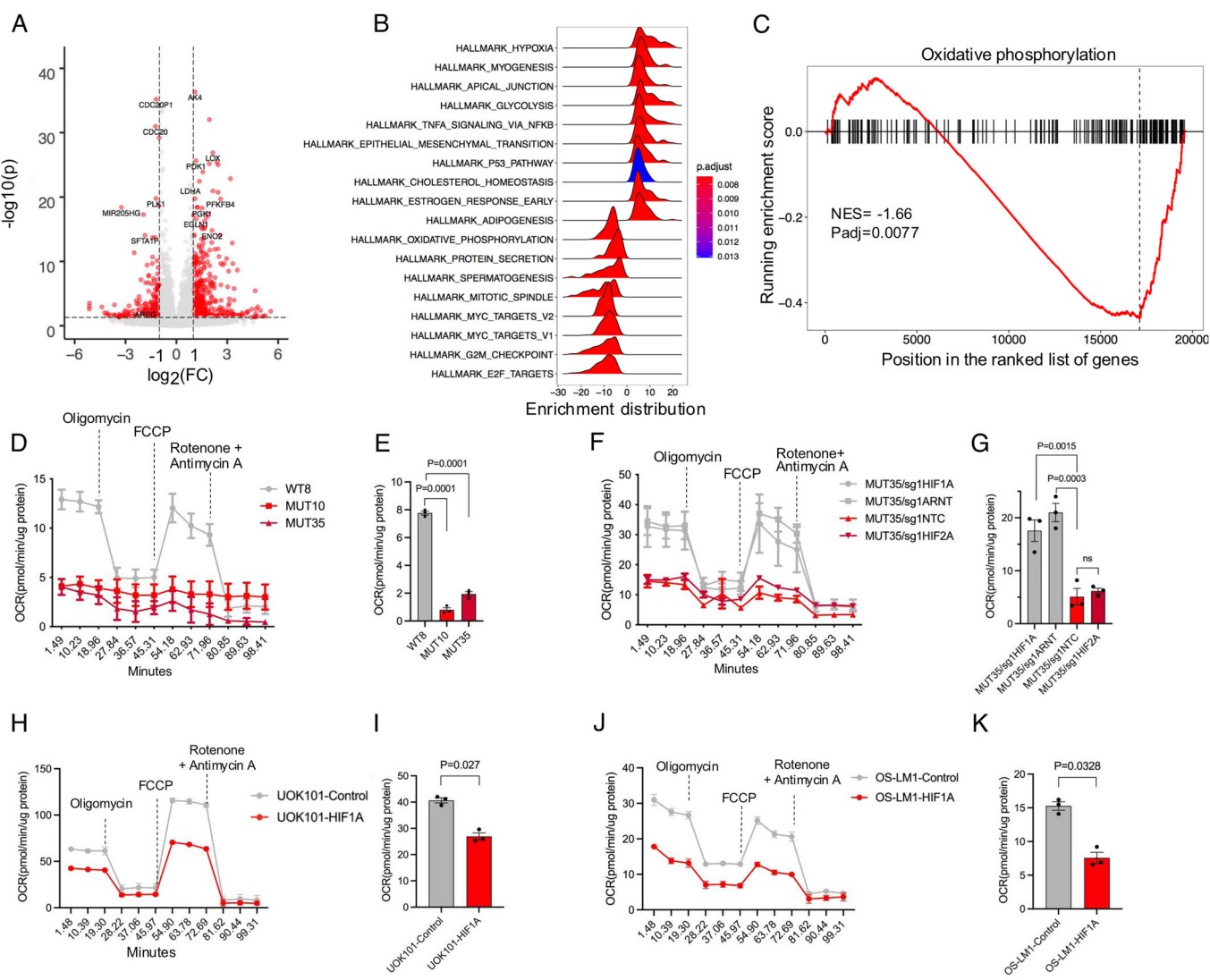

**Figure 2. HIF1A stabilization induces mitochondrial inhibition and glycolysis dependency in VHL mutant cells.**

(A) Differential gene expression analysis by RNA-seq. MUT10 and MUT35 cells compared to WT8 cells after dox withdrawal. *P* value and fold-change determined by DESeq2, using the Wald test to test the significance. *N* = 4 replicates per condition. (B) Gene set enrichment analysis (GSEA) comparing MUT10 and MUT35 cells to WT8 cells using the Cancer Hallmark gene set. (C) Running GSEA enrichment score for the oxidative phosphorylation gene set. (D) Oxygen consumption rate (OCR) in VHL WT and VHL MUT cells as measured by Seahorse analysis. *N* = 3 per condition (mean and S.E.M.). (E) Quantification of OCR value from mitochondria in VHL WT and VHL MUT cells. *N* = 3 per condition. Dunnett's multiple comparisons test (mean and S.E.M.). (F) Seahorse assay to measure Oxygen consumption rate (OCR) in MUT35-NTC, MUT35-sgHIF1A, MUT35-sgHIF2A and MUT35-sgHIF1B cells. *N* = 3 per condition (mean and S.E.M.). (G) Quantification of OCR value from mitochondria in MUT35-NTC, MUT35-sgHIF1A, MUT35-sgHIF2A and MUT35-sgHIF1B cells. *N* = 3 per condition. Sidak's multiple comparisons test (mean and S.E.M.). (H) Oxygen consumption rate (OCR) in UOK101 cells with and without HIF1A cDNA expression. *N* = 3 replicates per condition (mean and S.E.M.) as determined by Seahorse analysis. (I) OCR value in mitochondria in UOK101 cells with and without HIF1A cDNA expression. *N* = 3 replicates per condition (mean and S.E.M.). Student's *t* test. (J) Oxygen consumption rate (OCR) in OS-LM1 cells with and without HIF1A cDNA expression. *N* = 3 replicates per condition (mean and S.E.M.) as determined by Seahorse analysis. (K) OCR value in mitochondria in OS-LM1 with and without HIF1A cDNA expression. *N* = 3 replicates per condition (mean and S.E.M.). Student's *t* test. Source data are available online for this figure.

agreement with the cell proliferation data, HIF1A and ARNT inactivation rescued mitochondrial function and increased OCR value in *VHL* null cells (Fig. 2F,G). As previously demonstrated (Zhang et al, 2007; Shen et al, 2011), forced HIF1A expression also reduced proliferation and mitochondrial respiration in established ccRCC cell lines (Figs. 2H–K and EV4D,E). The HIF1A axis can thus reduce cell fitness in normal epithelial cells and advanced malignant cancer clones.

## HIF1A-dependent pharmacological vulnerabilities in *VHL* mutant cells

As HIF1A suppresses mitochondria, the main source of ATP, and as *VHL* null cells showed evidence of sensitivity to genes involved in glycolysis (Fig. EV3D), we tested the possibility of using glycolysis inhibitors to specifically target VHL null cells. However, 2-deoxyglucose (2-DG) and AZD3965, inhibitors of hexokinase

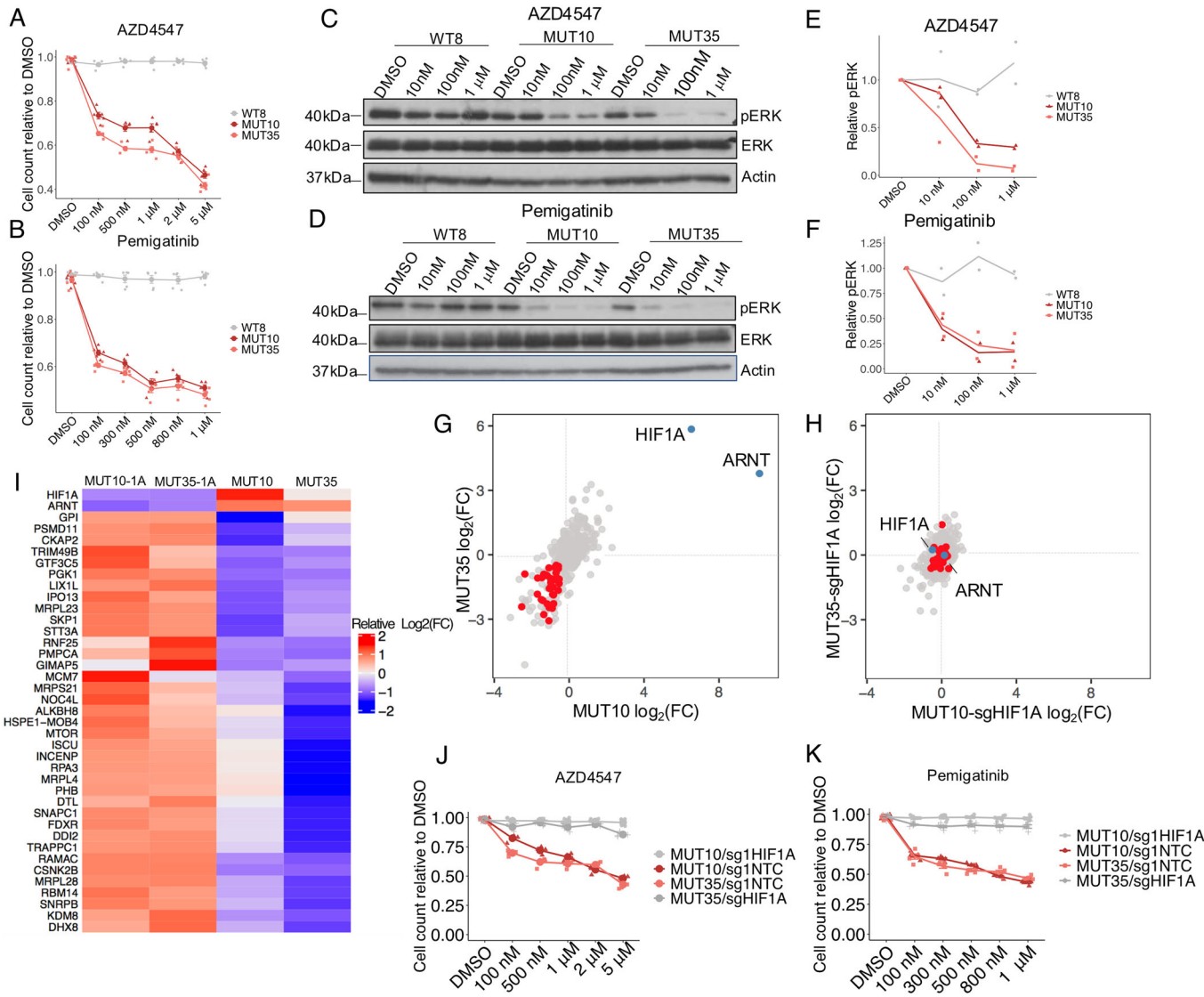

**Figure 3.  HIF1A dependent druggable genetic vulnerabilities in *VHL* mutant cells.**

(A, B) Cell counts on day 5 relative to DMSO control group under AZD4547 (100 nM, 500 nM, 1 μM, 2 μM, 5 μM) and pemigatinib (100 nM, 300 nM, 500 nM, 800 nM,1 μM) treatments. $N = 4$ per condition (Mean and SD). (C, D) Western blot of pERK, ERK and beta-actin in WT8, MUT10 and MUT35 cells treated with AZD4547 (DMSO, 10 nM, 100 nM, 1 μM) and Pemigatinib (DMSO, 10 nM, 100 nM, 1 μM). (E, F) Normalized pERK quantification in WT8, MUT10 and MUT35 cells treated with DMSO or AZD4547 (10 nM, 100 nM, 1 μM) or Pemigatinib (10 nM, 100 nM, 1 μM). $N = 2$ per condition (mean and S.E.M.). (G, H) A targeted CRISPR-Cas9-based loss of function pooled screen in MUT10 and MUT35 cells (G), and double mutant MUT10-sgHIF1A and MUT35-sgHIF1A cells (H). HIF1A dependent gene dependencies are labelled in red. (I) Heat map of 37 representative HIF1A dependent gene dependencies. Fold change in sgRNA abundance. (J, K) Cell counts on day 5 relative to DMSO control group under AZD4547 (100 nM, 500 nM, 1 μM, 2 μM, 5 μM) and Pemigatinib (100 nM, 300 nM, 500 nM, 800 nM,1 μM) treatments. $N = 3$-4 per condition (mean and SD). Source data are available online for this figure.

and monocarboxylate transporter 1/2, respectively, showed only a mild effect on *VHL* mutant cells (Fig. EV5A,B). To explore the possibility that VHL loss resulted in other pharmacological vulnerabilities, we looked for potential druggable VHL loss-induced genetic dependencies in our CRISPR-Cas9 screen data. This identified FGFR1 and RAD51 as potential targets for which small molecule inhibitors were available (Fig. EV5C). Pemigatinib and AZD4547 are kinase inhibitors with efficacy against FGFR1 (Gavine et al, 2012; Liu et al, 2020). Both inhibited the proliferation of MUT10 and MUT35 cells when compared to WT8 cells

(Fig. 3A,B). They also reduced ERK phosphorylation, a downstream effect of FGFR1 activation, more efficiently in MUT10 and MUT35 cells when compared to WT8 cells (Fig. 3C–F). B02, a RAD51 inhibitor (Huang et al, 2011), also inhibited the proliferation of MUT10 and MUT35 cells more efficiently when compared to WT8 cells (Fig. EV5D). Importantly, however, cisplatin (CDDP), doxorubicin, imatinib and navitoclax showed broadly similar effects on *VHL* mutant and wild-type cells, demonstrating specificity of our observations (Fig. EV5E–H). These results demonstrate that *VHL* mutations can induce pharmacological

vulnerabilities with associated biochemical phenotypes in renal epithelial cells.

The central role of HIF1A in mediating VHL loss-induced proliferative suppression suggested that the VHL loss-induced genetic and pharmacological vulnerabilities could in general be HIF1A dependent. We tested this by a secondary pooled CRISPR-Cas9 screen that targeted genes the sgRNAs of which were depleted (94 genes) or enriched (250 genes) in VHL mutant cells in the original genome-wide screen (Fig. EV6A,B). We also included constructs that targeted 30 genes frequently mutated in ccRCC and 22 non-essential control genes. The screen was performed using MUT10 and MUT35 cells together with the corresponding VHL-HIF1A double mutant cells MUT10-1A and MUT35-1A (Fig. EV6B). Overall, the secondary screen validated the original results from the genome-wide screen in the VHL mutant cells (Figs. 3G and EV6C). Interestingly, in the VHL-HIF1A double-mutant cells we observed little enrichment or depletion of sgRNAs, indicating that the genetic vulnerabilities identified in VHL mutant cells were HIF1A dependent in this context (Fig. 3H,I). Withdrawal of dox before lentiviral transduction of the sgRNA library also resulted in strong enrichment of HIF1A and ARNT constructs, indicating that the growth suppression caused by VHL loss and HIF1A stabilization was reversible (Fig. EV6D).

To further test whether the pharmacological vulnerabilities were also HIF1A dependent we treated the VHL mutant and VHL-HIF1A double mutant cells with pemigatinib, AZD4547 and B02. In all cases, VHL-HIF1A double mutant cells were significantly less sensitive when compared to VHL single mutant cells (Figs. 3J,K and EV6E). The increased sensitivity of VHL null cells to ERK inhibition was also HIF1A dependent (Fig. EV6F,G). These data demonstrate that VHL mutant cells, e.g., pre-cancerous VHL null clones in VHL mutation carriers, may have pharmacological vulnerabilities that are largely HIF1A-dependent and that could potentially be therapeutically targeted.

## Fitness advantage by VHL-HIF1A axis inactivation under PROTAC therapy

As shown above, the molecular consequences of VHL inactivation may be relevant for the development of novel intervention strategies for pathologies (e.g. ccRCC) that arise due to VHL inactivation. In addition, the VHL pathway may also be relevant for VHL wild-type cancers, because the E3 ubiquitin ligase activity of the VHL complex can be harnessed by PROTACs for targeted protein degradation (Fig. 4A). While recent evidence from functional screening suggests that several E3 proteins can be utilized for proximity-induced protein degradation (Poirson et al, 2022), the broadly essential role of VHL for the proliferation of non-ccRCC cancer cells (Fig. 1A) makes it attractive for PROTAC applications in cancer. Due to inactivating VHL mutations, ccRCCs are expected to be naturally resistant to VHL-dependent PRO-TACs, and in VHL wild-type cancers, VHL inactivation would also to lead to PROTAC resistance. However, because VHL loss leads to reduced proliferative fitness (Figs. 1A,B and EV1A–D), our results suggested that a proliferation proficient PROTAC resistant phenotype would only emerge through combined loss of VHL and HIF1A/ARNT. To explore the relevance of our observations for PROTAC therapies, we first tested the effect of ARV-771, a VHL-dependent bromodomain degrader that has previously shown anti-cancer activity (Raina et al, 2016), on a panel of cancer cell lines

representing several non-ccRCC cancers and the renal epithelial cell line HK2. All but HK2 cells were sensitive to ARV-771 (Fig. 4B). As predicted, the esophageal cancer cell line OE33, the breast cancer cell lines MCF7 and 1833-BoM, the prostate cancer cell line LNCaP and the myeloid leukemia cell line KBM7 became resistant to ARV-771 when VHL was inactivated (Fig. 4C). However, as HIF1A stabilization upon VHL suppression leads to reduced proliferative fitness, VHL single mutant cells were predicted not to be competitive in a population even if they were drug resistant, whereas VHL-HIF1A double mutant cells were expected to be resistant and able to proliferate. To test this experimentally, we mixed wild-type (mCherry labelled, 40%), VHL mutant (GFP labelled, 55%) and VHL-HIF1A double mutant (BFP labelled, 5%) 1833-BoM, MCF7 and OE33 cells for triple competition assays under DMSO or ARV-711 treatment (Figs. 4D and EV7A–C). The proliferation of WT, VHL mutant and VHL-HIF1A double mutant OE33, 1833-BoM and MCF7 cells was also tested separately (Fig. EV7D–F). In DMSO vehicle conditions for all cell lines, the wild-type cells had a proliferative advantage (Figs. 4E,F and EV7G–J). However, when the cells were treated with ARV-771, the abundance of wild-type cells reduced quickly (Figs. 4G,H and EV7K–N). The relative abundance of VHL mutant cells was briefly enriched, but eventually the VHL-HIF1A double mutant cells, even if present only as a small minority population at the start of the experiment, became the dominant population (Figs. 4G,H and EV7K–N).

To test whether the proliferative fitness advantage of VHL-HIF1A double mutant cells under ARV-771 treatment translated into an advantage over VHL mutants cells also in vivo, we mixed wild-type (BFP), VHL mutant (GFP) and VHL-HIF1A double mutant (GFP+mCherry) 1833-BoM cells, and then performed an in vivo competition assay in a context of ARV-771 treatment for 21 days (Fig. 5A). The in vivo xenograft tumor assay confirmed that ARV-771 reduced tumor growth (Fig. 5B), and immunohisto-chemical staining showed less Ki67 and more cleaved caspase 3 in ARV-771 treated tumors (Fig. 5C). Importantly, based on flow cytometry analysis, in comparison to untreated tumors, the relative abundance of VHL-HIF1A double mutant cells was increased, the abundance of VHL single mutant cells stayed unchanged, and the abundance of wild-type cells was decreased in ARV-771 treated tumors, indicating selective advantage of VHL-HIF1A double mutant cells under PROTAC therapy in vivo (Fig. 5D). These data demonstrate that inactivation of the VHL-HIF1A axis can cause functional resistance to VHL-dependent PROTACs in cancer cells.

## Uncoupling VHL functions in HIF1A regulation and PROTAC sensitivity

Genotype–phenotype analyses on human VHL mutation carriers suggest that some VHL variants predispose mainly to phaeochromocytoma and paraganglioma, while more severe alterations are associated with ccRCC (Minervini et al, 2019; Kaelin, 2022). HIF stabilization is typically associated with type 1 and type 2B mutations, while type 2A and 2C mutations retain the capability to target HIFs for degradation (Kaelin, 2022). This raised the possibility that the ability of VHL to mediate PROTAC effects could also be uncoupled from its role in HIF1A regulation. Consequently, there could be VHL mutations that disrupt PROTAC activity but do not lead to HIF1A stabilization and reduced proliferative fitness. To test this, we examined data from recent

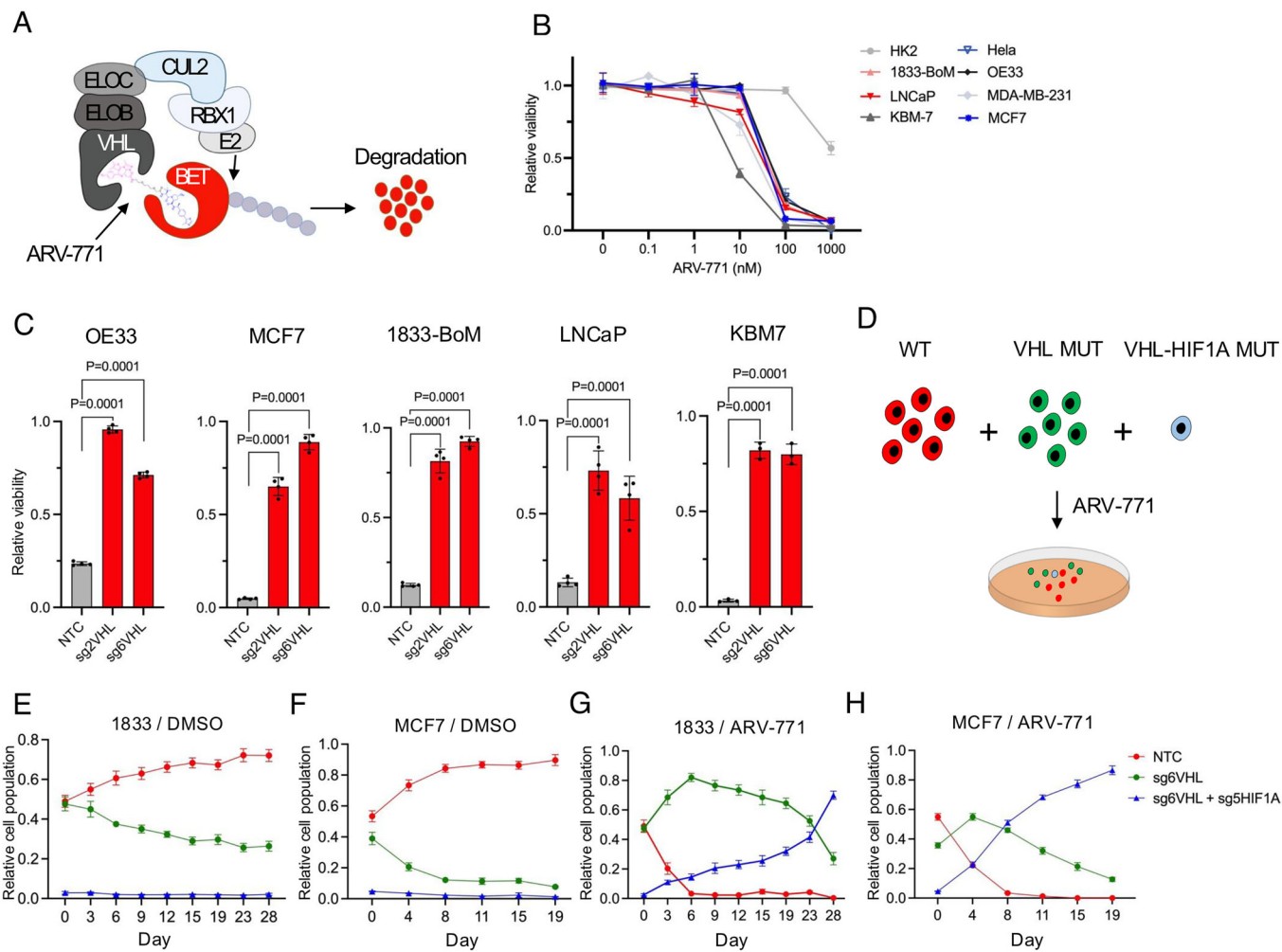

**Figure 4. PROTAC resistance through HIF1A inactivation in cancer.**

(A) Schematic of ARV-771 function as a PROTAC BET degrader. (B) Relative viability of cells upon ARV-771 treatment ($N = 3$ replicates per condition; error bar, SD; Dunnett's multiple comparisons test). (C) Relative cell viability upon ARV-771 treatment in wild-type and VHL mutant cells. ARV-771 concentrations as follows: OE33 (100 nM), MCF7 (1 μM), 1833-BoM (100 nM), LNCaP (100 nM), KBM7 (1 μM). $N = 3$–4 replicates per condition (mean and S.E.M.). Dunnett's multiple comparisons test. (D) Schematic of long-term competition assay of WT (mCherry), VHL-MUT (GFP) and VHL-HIF1A double mutant (BFP) 1833-BoM, OE33 and MCF7 cells, treated with DMSO or ARV-771 (500 nM for 1833-BoM, 400 nM for OE33, 200 nM for MCF7). (E–H) FACS-based quantification of the relative abundances of different cell populations in competition assays. WT (mCherry), VHL-MUT by VHLsg6 (GFP) and VHL-HIF1A double mutant by VHLsg6 and HIF1Asg5 (BFP). 1833-BoM cells: DMSO (E) and MCF7 cells: DMSO (F). 1833-BoM cells: 500 nM ARV-771 (G) and MCF7 cells: 200 nM ARV-771 (H). $N = 3$ for each condition and timepoint (mean and SD). Source data are available online for this figure.

*VHL* saturation mutagenesis screens (Hanzl et al, 2023; Buckley et al, 2024) to identify amino acid residues that were associated with PROTAC insensitivity and VHL loss-induced proliferation phenotypes. Point mutants of *VHL* residues Y98 and P99 emerged as potential candidates. We generated dox-inducible empty vector, $VHL^{WT}$, $VHL^{Y98N}$, $VHL^{Y98F}$, $VHL^{Y98H}$, $VHL^{P99G}$ and $VHL^{P99M}$ expression constructs and transduced them into *VHL* knock-out MCF7 cells. Based on immunoblot analysis, only $VHL^{WT}$, $VHL^{Y98F}$ and $VHL^{P99M}$ were able to reduce HIF1A expression (Fig. 5E). In line with this, only cells expressing $VHL^{Y98F}$ and $VHL^{P99M}$ were able to proliferate at the same rate as $VHL^{WT}$ cells (Fig. 5F). When cells were exposed to ARV-771, while $VHL^{WT}$ and $VHL^{P99M}$ showed sensitivity to ARV-771, cells with every other construct including $VHL^{Y98F}$ showed only limited sensitivity (Fig. 5G). Thus, $VHL^{Y98F}$ was able to abrogate PROTAC sensitivity

while maintaining proliferative capacity at the level of $VHL^{WT}$ cells (Fig. 5H). Altogether, our results show that HIF1A stabilization provides potential opportunities for early intervention in pre-neoplastic *VHL* clones. In addition, we propose that the VHL-HIF1A axis may be relevant for the development of proliferation-proficient therapy resistance to the emerging PROTAC-based anti-cancer strategies.

# Discussion

The VHL-HIF1A pathway regulates metazoan responses to reduced oxygen availability, and the phenotypic sequelae of VHL inactivation are relevant in several clinical contexts ranging from carcinogenesis (Kaelin, 2007) and metabolic dysfunction (Perrotta

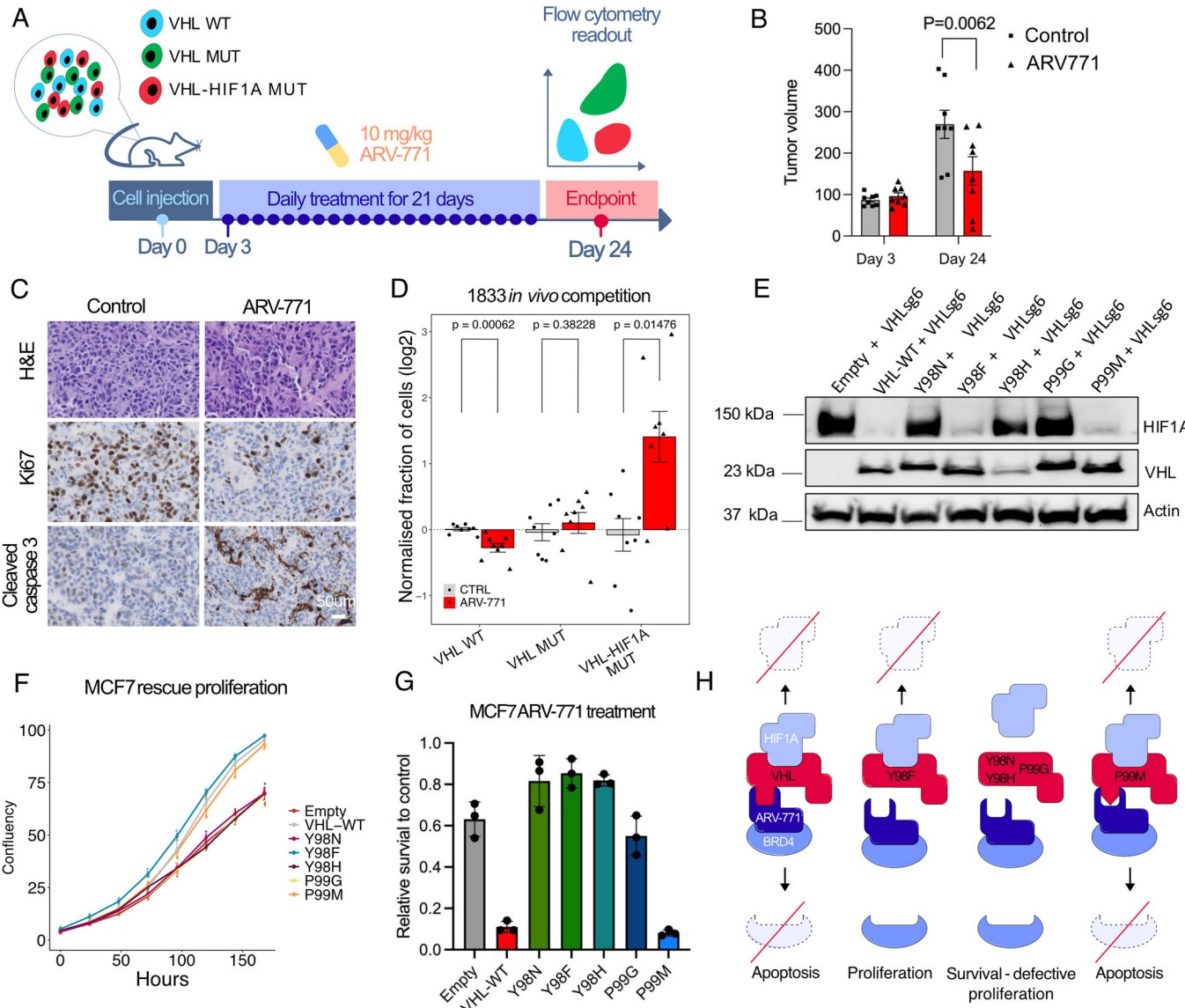

**Figure 5. Uncoupling VHL functions in HIF1A regulation and PROTAC sensitivity.**

(A) Schematic of in vivo competition assay. Two million WT (BFP), VHL-MUT (GFP) and VHL-HIF1A double mutant (mCherry+GFP) 1833-BoM cells subcutaneously inoculated into the flanks of athymic nude mice. Mice were treated daily with DMSO or 10 mg/kg ARV-771 for 21 days. Tumors were dissociated for FACS analysis. (B) Tumor size on day 3 and day 24. $N = 8$ tumors for each condition (mean and S.E.M.). Sidak's multiple comparisons test. (C) H&E, Ki67 and Cleaved Caspase 3 staining of tumors treated with DMSO and ARV-771. (D) Log2 of relative cell population for each subpopulation of indicated treatment. $N = 8$ tumors for each condition (mean and S.E.M.). Wilcoxon test. (E) HIF1A and VHL western blot on empty vector, VHL$^{WT}$, VHL$^{Y98N}$, VHL$^{Y98F}$, VHL$^{Y98H}$, VHL$^{P99G}$ and VHL$^{P99M}$ expressing VHL mutant MCF7. (F) Incucytes proliferation of VHL-mutant MCF7 while expressing with empty vector, VHL$^{WT}$, VHL$^{Y98N}$, VHL$^{Y98F}$, VHL$^{Y98H}$, VHL$^{P99G}$ and VHL$^{P99M}$. $N = 3$ replicates per condition. (G) Relative cell viability upon 1 μM ARV-771 treatment in empty vector, VHL$^{WT}$, VHL$^{Y98N}$, VHL$^{Y98F}$, VHL$^{Y98H}$, VHL$^{P99G}$ and VHL$^{P99M}$ expressing VHL-KO MCF7 cells. $N = 3$. (mean and S.E.M.). (H) Schematic of VHLY98F resistant to ARV-771 treatment and not activate HIF1A protein. Source data are available online for this figure.

et al, 2020) to novel cancer therapies (Békés et al, 2022). Using unbiased functional screening in experimental systems, we have thus characterized the consequences of acute VHL loss, demonstrating that HIF1A activation in VHL null cells can lead to potentially actionable genetic vulnerabilities.

Our data suggest that HIF1A is the central mediator of VHL loss-induced negative fitness effects, with little involvement of HIF1A-independent pathways detected in our screen. This is in line with recent data on mouse embryonic fibroblasts (Hoefflin et al, 2020), but earlier studies have also reported HIF1A-independent mechanisms through which VHL loss may inhibit cell proliferation (Young et al, 2008; Welford et al, 2010). Interestingly, recent data suggest that in some cell types VHL loss-induced activation of HIF2A may be the critical effector of fitness loss, pointing at cell type-specific effects (Abu-Remaileh et al, 2024). We show that VHL loss suppresses mitochondrial function, a result concordant with known effects of

HIF1A on mitochondrial activity (Papandreou et al, 2006; Kim et al, 2006), but the detailed mechanisms remain to be elucidated in future studies. Importantly, we demonstrate that HIF1A activation in VHL null cells can result in pharmacologically targetable genetic vulnerabilities, such as increased sensitivity to FGFR1 or RAD51 inhibition. These results provide a proof of principle that non-cancerous VHL-null cells can be susceptible to therapeutic intervention, complementing VHL synthetic lethal screens previously performed in already established VHL mutant renal cancer cell lines (Sun et al, 2019; Bertlin et al, 2024). For example, the cancer risk of individuals carrying heterozygous VHL mutations could be reduced even with incomplete early eradication of premalignant VHL-null clones. Expanding the concepts from our findings into in vivo models in which VHL-null cells could be analyzed in a non-proliferative state would be a critical next step. On the other hand, HIF1A-dependent suppression of mitochondrial activity in non-cancerous VHL-null cells could explain the systemic metabolic defects associated with biallelic germline mutations in VHL (Perrotta et al, 2020). Strategies to inhibit HIF1A activity could ameliorate the severe phenotype in such patients.

PROTACs are an emerging class of versatile molecules that can harness endogenous E3 ubiquitin ligase activity for specific degradation of target proteins of interest. Most PROTACs in clinical and pre-clinical development use CRBN or VHL as the endogenous E3 ubiquitin ligase proteins (Békés et al, 2022). Loss of the relevant E3 complex compromises PROTAC activity (Mayor-Ruiz et al, 2019), but as VHL is broadly essential for cancer cell proliferation, VHL loss alone is unlikely to result in clinically relevant PROTAC resistance. Our data suggest that cell proliferation-proficient resistance to VHL-dependent PROTACs, intrinsically rare compared to resistance to CRBN-dependent PROTACS (Hanzl et al, 2023), can result from combined VHL and HIF1A inactivation, or through VHL mutations that retain the activity to target HIF1A for degradation. Reducing the probability of HIF1A inactivation in VHL deficient clones under therapy could improve long-term efficacy of PROTAC treatment. This could be achieved through the elimination of HIF1A-positive VHL-null cells before they lose HIF1A, possibly by targeting HIF1A-induced vulnerabilities as demonstrated by our data. Alternatively, the identification of synthetic lethal approaches for HIF1A null cells could be beneficial.

In conclusion, our data provide genome-wide functional insight into the molecular consequences of VHL loss in human cells. This revealed specific, mostly HIF1A-dependent genetic vulnerabilities in VHL null cells, suggesting the possibility that VHL loss-induced phenotypes could be exploited therapeutically. We also describe mechanisms that can provide cancer cells with proliferative fitness upon resistance to VHL-dependent PROTACs. This could aid the future development and clinical applicability of this promising novel class of anti-cancer agents.

# Methods

### Reagents and tools table

| Reagent/resource | Reference or source | Identifier or catalog number |
|---|---|---|
| RIPA lysis buffer | Merck Life Science Limited | R0278-50ML |
| **Experimental models** | | |
| HK2 (*H. sapiens*) | C. Frezza (MRC Cancer Unit) | N/A |
| KBM7 (*H. sapiens*) | A. Obenauf (IMP, Vienna) | N/A |

| Reagent/resource | Reference or source | Identifier or catalog number |
|---|---|---|
| OE33 (*H. sapiens*) | Rebecca Fitzgerald (MRC Cancer Unit) | N/A |
| LnCAP (*H. sapiens*) | Charlie Massie (Early Cancer Institute) | N/A |
| UOK101 (*H. sapiens*) | M. Linehan (NCI, Bethesda) | N/A |
| OS-LM1 (*H. sapiens*) | J. Massagué (MSKCC, New York) | N/A |
| 1833-BoM (*H. sapiens*) | J. Massagué (MSKCC, New York) | N/A |
| 293 T (*H. sapiens*) | J. Massagué (MSKCC, New York) | N/A |
| Human renal organoids (*H. sapiens*) | Addenbrokes Hospital | N/A |
| MDA-MB-231 (*H. sapiens*) | J. Massagué (MSKCC, New York) | N/A |
| Athymic nude (*Mus musculus*) | Charles River | N/A |
| **Recombinant DNA** | | |
| psPAX2 | Addgene | # 12260 |
| pMD2.G | Addgene | # 12259 |
| Lenti-V2 | Addgene | # 52961 |
| Lenti-cas9-blast | Addgene | # 52962 |
| LT3-GEPIR | Addgene | #111177 |
| LT3-VHL-GFP-hygro | Cloned by lab | N/A |
| LT3-VHL$^{Y98N}$-GFP-hygro | Cloned by lab | N/A |
| LT3-VHL$^{Y98F}$-GFP-hygro | Cloned by lab | N/A |
| LT3-VHL$^{Y98H}$-GFP-hygro | Cloned by lab | N/A |
| LT3-VHL$^{P99G}$-GFP-hygro | Cloned by lab | N/A |
| LT3-VHL$^{P99M}$-GFP-hygro | Cloned by lab | N/A |
| pKLV2-BFP-Puro-U6-sgRNA | Addgene | #67974 |
| pKLV2-GFP-hygro-U6-sgRNA | Cloned by lab | N/A |
| pKLV2-cherry-hygro-U6-sgRNA | Cloned by lab | N/A |
| **Antibodies** | | |
| VHL | BD | 565183 |
| HIF1A | Proteintech | 20960-1-AP |
| HIF2A | Novus Biologicals | NB100-122 |
| actin | Sigma-Aldrich | A1978 |
| p-ERK | Abcam | ab201015 |
| ERK | Abcam | ab184699 |
| Cleaved Caspase-3 | Cell signaling | 9661S |
| Ki67 | Abcam | Ab16667 |
| polyclonal goat anti-mouse IgG/HRP | Dako | P0447 |

| Reagent/resource | Reference or source | Identifier or catalog number |
|---|---|---|
| goat anti-rabbit IgG/ HRP | Dako | P0448 |
| **Oligonucleotides and other sequence-based reagents** | | |
| **Chemicals, enzymes and other reagents** | | |
| RPMI-1640 | Thermo Fisher | 11875093 |
| Fetal bovine serum | Thermo Fisher Scientific | A5256701 |
| DMEM high glucose | Gibco | 41965-039 |
| Agilent Seahorse XF Calibrant | Agilent Technologies | #100840-000 |
| Linear polyethylenimine (PEI), 25 kDa | Polysciences | 23966-2 |
| Polybrene | Millipore | TR-1003-G |
| Hygromycin | Invitrogen | 10687010 |
| Puromycin | GIBCO | A1113803 |
| Blasticidin S HCl | GIBCO | A1113903 |
| RIPA lysis buffer | Merck Life Science Limited | R0278-50ML |
| DPBS | ThemoFisher | 14190169 |
| Pemigatinib | MCE | HY-109099 |
| AZD4567 | Apex Bio Tech | A8350 |
| 2-DG | Universal Biological Ltd | S4701-100mg |
| AZD3965 | Cambridge Bioscience | HY-12750-5mg |
| CDDP | Cambridge Bioscience | HY-17394-50mg |
| Imatinib | Cambridge Bioscience | T6230-100mg |
| Navitoclax | Cambridge Bioscience | HY-10087-5mg |
| Doxorubicin | Fluorochem Limited | F021790-250mg |
| B02 | TOCRIS | 1 A/246295 |
| ARV771 | MCE | #HY-100972 |
| FCCP | Tokoyo Chemical Industry | c3453-10mg |
| Oligomycin | Cambridge Bioscience | o4533-1mg |
| **Software** | | |
| GraphPad Prism 10 | https://www.graphpad.com/ updates/prism-1000-release-notes | |
| ImageJ | https://imagej.net/ij/ | |
| **Other** | | |
| Incucyte software (v2020C) | Sartorius | |
| Tecan Infinite M1000Pro | Tecan | |
| Olympus microscopic | Olympus | |

## VHL dependency in the DepMap database

VHL dependency in ccRCC cell lines vs other cancer cell lines was calculated using the Cancer Dependency Map CRISPR/Cas9 loss-of-function screening data set, 25Q2 release (Arafeh et al, 2025). To determine ccRCC identity, a manually curated list of ccRCC cell lines was used (provided as source data). The Wilcoxon test was used to calculate significance.

## Cell lines

The HK2 cell line was obtained from C. Frezza (MRC Cancer Unit, Cambridge, UK). KBM7 cells were obtained from A. Obenauf (IMP, Vienna). OE33 cells were obtained from Rebecca Fitzgerald (MRC Cancer Unit, Cambridge, UK). LnCAP cells were obtained from Charlie Massie (Early Cancer Institute, Cambridge, UK). UOK101 cells were obtained from M. Linehan (NCI, Bethesda). OS-LM1 (Vanharanta et al, 2013), MDA-MB-231, 1833-BoM (Kang et al, 2003) and 293T cells were obtained from J. Massagué (MSKCC, New York, USA). WT8, MUT10 and MUT35 are single cell clonal derivatives of HK2 cells with VHL mutation generated by CRISPR-Cas9 using VHLsg6. VHL inactivation was validated by Sanger sequencing and Western blotting for VHL and HIF2A. The identity of the cell lines was confirmed by STR analysis. Cells were also confirmed to be mycoplasma free. HK2 and ccRCC cell lines were cultured in RPMI-1640 medium (Sigma) supplemented with 10% FBS, penicillin (100 U/mL) and streptomycin (100μg/mL). All other cell lines were cultured in high glucose DMEM medium (Invitrogen) supplemented with 10% FBS, penicillin (100 U/mL) and streptomycin (100μg/mL).

## Plasmids

psPAX2 and pMD2.G were gifts from Didier Trono (Addgene # 12260 and # 12259), lenti-cas9-blast was a gift from Feng Zhang (Addgene# 52962). pKLV2-U6gRNA5(BbsI)-PGKpuro2ABFP-W (Addgene#67974) was a gift from Kosuke Yusa. pKLV2-U6gRNA5(BbsI)-PGKhygro2ABFP, pKLV2-U6gRNA5(BbsI)-PGKpuro2AGFP and pKLV2-U6gRNA5(BbsI)-PGKpuro2Am-Cherry are derivatives of pKLV2-U6gRNA5(BbsI)-PGKpuro2ABFP-W. The dox-inducible shRNA expression plasmid LT3GEPIR was kindly gifted by J. Zuber (IMP, Vienna). sgRNA sequences are listed in Dataset EV4. HIF1A cDNA was amplified from HA-Clover-HIF-1alpha Wild-type (Addgene#163365). pLVX-puro (632164, Clonetech) was used to exogenously express the cDNA constructs. pLVX-HRE-ODD-GFP-hygro was generated by fusing GFP with HIF1A-ODD domain and cloned under a hypoxia response element modified minimal CMV promoter.

## Lentiviral production and transduction

HEK293T cells were transfected with a mixture of the lentiviral transfer plasmid containing genes of interest, psPAX2 and pMD2.G using PEI reagent (MW.25,000. Alfa Aesar). The supernatant containing the lentivirus was collected 72 h post-transfection and filtered by a 0.45-μM PVDF sterile filter (ELKAY). Cells were transduced with the lentiviral supernatant in the presence of 5 μg/mL polybrene (Millipore). Puromycin (4 μg/mL), hygromycin (800 μg/mL, InvivoGen) or blasticidin (10 μg/mL, InvivoGen). Selection started 48 h post-transduction.

## Pooled CRISPR-Cas9 screening

The lentiviral genome wide sgRNA library was produced using HEK293T cells as described above. A total of 450 million cells per

condition were transduced with the lentiviral library at a low MOI (<0.3) to ensure that 95% of cells had a single sgRNA integration, resulting in 500× sgRNA representation. Puromycin was added 48 h post-infection. Dox was removed from culture media 7 days post-infection to stop transgene VHL expression. Pooled cells were allowed to proliferate for 4 weeks before harvested for DNA extraction. Genomic DNA was extracted using the QIAamp DNA Blood Maxi Kit (Qiagen Cat # 51192). All DNA was used for the amplification of the sgRNA cassette. Amplified product was purified by 1% agarose gel, quantified with the Qubit dsDNA HS assay kit (Thermo) and pooled in equimolar concentrations prior to Illumina sequencing on a HiSeq4000 instrument. Sequencing results were analyzed by the MAGeCK protocol (Wang et al, 2019).

## In vitro proliferation assays

For proliferation phenotype assays, all cells were plated on a 24-well plate in triplicates with a start confluency ranging from 10 to 15%. Proliferation was measured by Incucyte 2020. For competition assays, control and target cells, which carried different fluorescent markers (BFP + /mCherry + /GFP + ), were mixed and plated onto multi-well plates in triplicates. The percentage of each cell population was analyzed from day 0 and multiple time points throughout the assays by flow cytometry on LSR Fortessa (BD Biosciences). The following gating approach was used: FSC-A, FSC-W, SSC-A to select for live and single cells, and then BFP (383 nm/445 nm), mCherry (561 nm/610 nm), or GFP (488 nm/510 nm) channels for discriminating between the cell populations.

## Protein detection

Total protein was extracted from cell pellets using RIPA buffer (Sigma) containing protease K and phosphatase inhibitor cocktail (Sigma) according to the manufacturer's protocol. Proteins were separated by SDS-PAGE, transferred onto PVDF membrane (Millipore) and blotted with VHL (BD Pharmingen, 565183, 1:500), HIF1A (Proteintech, 20960-1-AP, 1:1000), HIF2A (Novus Biologicals, NB100-122, 1:1000), beta-actin (Sigma-Aldrich, A1978, 1:30,000), p-ERK (Abcam, ab201015, 1:1000), ERK (Abcam, ab184699, 1:2000) antibodies. Secondary antibodies were poly-clonal goat anti-mouse IgG/HRP (Dako, P0447, 1:10,000) and polyclonal goat anti-rabbit IgG/HRP conjugated (Dako, P0448, 1:5000). The membranes were stripped between blottings. Protein expression was quantified using ImageJ and normalized to beta-actin.

## RNA-seq

RNeasy Mini Kit (Qiagen 74104) was used for total RNA extraction on sub-confluent cells in four replicates according to the manufacturer's protocols. The quality and concentration were assessed with the Agilent RNA Nano 6000 kit (Agilent 5067-1511) on Agilent Bioanalyzer 2100 instrument. RNA-seq libraries were prepared using the SENSE/CORALL mRNA-seq Library Prep Kit (Lexogen), 1 µg of total RNA was used as the starting material following the manufacturer's recommendations. The library size and quality of the final library products were assessed using Agilent High Sensitivity DNA Kit (Agilent 5067-4626). Library concentration was determined using the KAPA Library Quantification Kit

(KR0405). Libraries were pooled in equimolar concentrations and subjected to Illumina sequencing on a HiSeq4000 instrument. Single clone RNA-seq sequencing reads were mapped to hg38 using RSEM and bowtie2. Differentially expressed genes were identified using DeSeq2 (Love et al, 2014). Gene set enrichment analysis was performed using R packages ClusterProfiler (Yu et al, 2012) and Molecular Signature Database (MSigDB) Hallmarks gene set (Version 7.1.1).

## Seahorse experiments

To assess the oxygen consumption rate (OCR) 80,000 cells were seed in XF$^e$ 24 well Cell Culture microplate in 100 µL normal RPMI night before experiment. The next day cells were washed once in PBS and the medium was replaced with 675 µL of XF RPMI Medium (Agilent Seahorse, 103576-100) supplemented with 25 mM glucose, 1 mM pyruvate, 4 mM glutamine. To eliminate residues of carbonic acid from medium, cells were incubated for at least 30 min at 37 °C with atmospheric $CO_2$ incubator. OCR was assayed in a Seahorse XF-24 extracellular flux analyzer by the addition via ports A-C of 1 µM oligomycin (port A), 1 µM carbonyl cyanide-p-trifluoromethoxyphenylhydrazone (FCCP, port B), 1 µM antimycin A (port C). Two or three measurement cycles of 2 min mix, 2 min wait, and 4 min measure were carried out at basal condition and after each injection. Each well was washed twice with 1 mL PBS and proteins were extracted with 50 µL of radioimmune precipitation assay (RIPA) lysis medium at room temperature end of the experiment. Protein concentration in each well was measured by a BCA assay according to the manufacturer's instructions (Thermo). OCR values were normalized to total µg of proteins in each well.

## Human renal epithelial organoids

Human renal epithelial organoids were generated using a previously described method (Schutgens et al, 2019). Normal human kidney tissue was sampled with informed consent by a consultant uropathologist within 2 h of nephrectomy under an ethical approval by the East of England - Cambridge Central Research Ethics Committee (19/EE/0161). The tissue was trans-ported on ice in cold HBSS, dissected in PBS supplemented with penicillin (100 U/mL)/streptomycin (100µg/mL) and further split into advanced DMEM/F12 (Gibco). Kidney tissue pieces were minced, washed in wash medium (advanced DMEM/F12 supplemented with 1× Glutamax, penicillin (100 U/mL), streptomycin (100µg/mL) and 10 mM HEPES) and resuspended in kidney organoid medium containing collagenase A (1 mg/mL, Sigma) for 45 min at 37 °C with shaking. The cells were washed, pelleted by centrifugation (5 min, 300 rcf, 4 °C), resuspended in wash medium, passed through a 70-µm strainer, pelleted by centrifugation (5 min, 300 rcf, 4 °C) and resuspended in 100 µl wash medium. Single cells were seeded in 70% growth factor-reduced BME (R&D Systems) and cast into 20 µl droplets in a 12-well plate. After polymerization of the BME (30 min, 37 °C), 1 mL kidney organoid medium was added to each well. For passage, organoids were dissociated using 1 mL TrypLE (Gibco) containing 10 µM Y-27632 per well (InvivoGen). TrypLE dissociation was stopped by adding 10 mL advanced DMEM/F12 and centrifuged at 300 rcf for 5 min, cells were reseeded in fresh 70% BME and topped

## The paper explained

### Problem

The von Hippel-Lindau tumor suppressor (*VHL*) controls cellular responses to hypoxia. *VHL* inactivation has many clinically relevant consequences: biallelic *VHL* inactivation in the kidney leads to the development of clear cell renal cell carcinomas (ccRCCs), *VHL* mutations in the germline lead to a tumor predisposition syndrome characterized by the development of different tumor types, and biallelic *VHL* inactivation in the germline can result in a severe metabolic disorder. Also, *VHL* wild-type cancers—the majority except for ccRCC—can be targeted by VHL-dependent proteolysis-targeting chimera (PROTAC) protein degraders, suggesting that VHL loss could mediate resistance mechanisms to PROTAC therapies.

Despite the relevance of VHL for several independent clinical contexts, the functional consequences of VHL inactivation remain incompletely understood. Here, we use experimental cell line systems and CRISPR-Cas9 functional screening to investigate the consequences of VHL loss at the genome-wide scale.

### Results

We show that the proliferation deficiency of VHL-null renal epithelial cells is mediated by HIF1A-dependent suppression of mitochondrial function. We have also identified new genetic and pharmacological vulnerabilities induced by VHL inactivation. Because HIF1A inactivation rescues the proliferation deficiency of VHL-null cells, we hypothesized that the combined inactivation of VHL-HIF1A could lead to proliferation-proficient PROTAC resistance. In line with this, we demonstrate that VHL-HIF1A null cells show a proliferative advantage compared to wild-type and VHL-null cells under treatment with the PROTAC drug ARV-771 in vitro and in vivo, indicating that inactivation of the VHL-HIF1A axis can result in functionally significant resistance to VHL-dependent PROTACs.

### Impact

VHL inactivation has important consequences in many clinical contexts, including carcinogenesis, metabolic dysfunction and cancer therapies. We show a proof-of-principle that VHL null cells have specific pharmacologically targetable genetic vulnerabilities, raising the possibility that new molecularly targeted intervention strategies could be developed for individuals carrying *VHL* mutations. We also show that the combined loss of VHL and HIF1A function is a possible mechanism of resistance to VHL-dependent PROTAC therapies, pointing to new strategies to improve the development and clinical applicability of this new class of anti-cancer agents.

with kidney organoid medium: ADMEM/F12 supplemented with 1.5% B27 supplement (Gibco, 17504044), 40% Wnt3A conditioned medium, 10% RSPO-conditioned medium, EGF (50 ng/mL, Proteintech), FGF-10 (100 ng/mL, Proteintech), N-acetylcysteine (1.25 mM/mL, Sigma), Y-27632 (10 μM, Apebio) and primocin (100 ng/mL, InvivoGen). The experiments conformed to the principles set out in the WMA Declaration of Helsinki and the Department of Health and Human Services Belmont Report.

### Organoid lentiviral transduction

Lentiviral supernatant was collected and mixed with LentiX concentrator (TaKaRa) in a ratio of 3:1. The mixture was incubated for 30 min at 4 °C, and centrifuged at 4 °C (1500 rcf, 1 h). Virus pellets were resuspended in renal organoid medium with polybrene (5 μg/ml). Organoids were dissociated into single cells and organoid pellets were resuspended in concentrated virus in 15 ml Falcon tubes and incubated for 4 h as previously described (Koo et al,

2011). The falcon tubes were then centrifuged at 600 rcf at 32 °C for 1 h before reseeding organoids in fresh 70% BME.

## Statistical analyses

Statistical analyses were performed either in R or GraphPad Prism. $P$ values lower than 0.05 were considered statistically significant. For correlation analyses, Pearson correlation coefficient was calculated. For drug treatment assays, two-way ANOVA with Sidak's multiple comparisons test was used. For seahorse experiments, ATP quantification analysis, ordinary one-way ANOVA with Sidak's multiple comparisons test was used.

## Animal studies

The animal experiment was performed in accordance with protocols approved by the Home Office (UK) and the University of Cambridge Animal Welfare and Ethical Review Body (P7EC604EE). Mice were housed under a 12 h light/dark cycle in a temperature (20–24 °C) and humidity (45–65%) controlled facility. No blinding, sample size estimation or randomization was used. No data were excluded. For subcutaneous tumor growth assays, $3 \times 10^6$ cells in 100 μL of 1:1 PBS/Matrigel Matrix (BD) solution were injected into both flanks of 5–6-week-old female athymic nude mice (Charles River Laboratories). Tumor growth was followed by caliper measurements on day 3 and day 24. Tumor volume (V) was calculated using the equation $V = (\text{length} \times \text{width}^2) \times 0.5$. ARV-771 treatment started on day 3 following cell transplantation and was given daily for 21 days.

## Data availability

The RNA-seq data generated within this project have been uploaded into the Gene Expression Omnibus under the access code GSE213241. The Cancer Dependency Map CRISPR/Cas9 loss-of-function screening dataset (25Q2 release) is publicly available on the DepMap portal at https://depmap.org/portal.

The source data of this paper are collected in the following database record: biostudies:S-SCDT-10_1038-S44321-025-00361-w.

## Peer review information

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

## Acknowledgements

We thank M Linehan for the UOK101 cells. GDS is supported by the Mark Foundation for Cancer Research and the Cancer Research UK Cambridge Centre (C9685/A25177). GDS and the Human Research Tissue Bank were supported by the NIHR Cambridge Biomedical Research Centre (BRC-1215-20014). The views expressed are those of the author(s) and not necessarily those of the NIHR or the Department of Health and Social Care. SH received a PhD studentship from the Rosetrees Trust. This project has received funding from the European Union's Horizon 2020 research and innovation program under the Marie Skłodowska–Curie grant agreement No 955951. This work was supported by the Medical Research Council (MC_UU_12022/7), Kidney Research UK (RP_033_20170303), the Sigrid Jusélius Foundation, the Academy of Finland (decision 338420) and the Cancer Foundation Finland. Munoz-Espin laboratory is supported by a CRUK Programme Foundation Award (C62187/A29760). JG is funded by a Darley/Sands Downing College Fellowship (G109261).

## Author contributions

**Jianfeng Ge**: Conceptualization; Data curation; Formal analysis; Validation; Investigation; Writing—original draft; Writing—review and editing. **Shoko Hirosue**: Formal analysis; Visualization. **Leticia Castillon**: Formal analysis. **Saroor A Patel**: Formal analysis. **Ludovic Wesolowski**: Formal analysis. **Anna Dyas**: Investigation. **Cissy Yong**: Resources; Provided human material. **Sanne de Haan**: Investigation; Assisted with the organoid experiments. **Jarno Drost**: Investigation; Assisted with the organoid experiments. **Grant D Stewart**: Resources; Provided human material. **Anna C Obenauf**: Conceptualization; Formal analysis. **Daniel Muñoz-Espín**: Supervision; Funding acquisition. **Sakari Vanharanta**: Conceptualization; Formal analysis; Supervision; Funding acquisition; Writing—original draft; Writing—review and editing.

Source data underlying figure panels in this paper may have individual authorship assigned. Where available, figure panel/source data authorship is listed in the following database record: biostudies:S-SCDT-10_1038-S44321-025-00361-w.

## Disclosure and competing interests statement

The authors declare no competing interests.

# Expanded View Figures

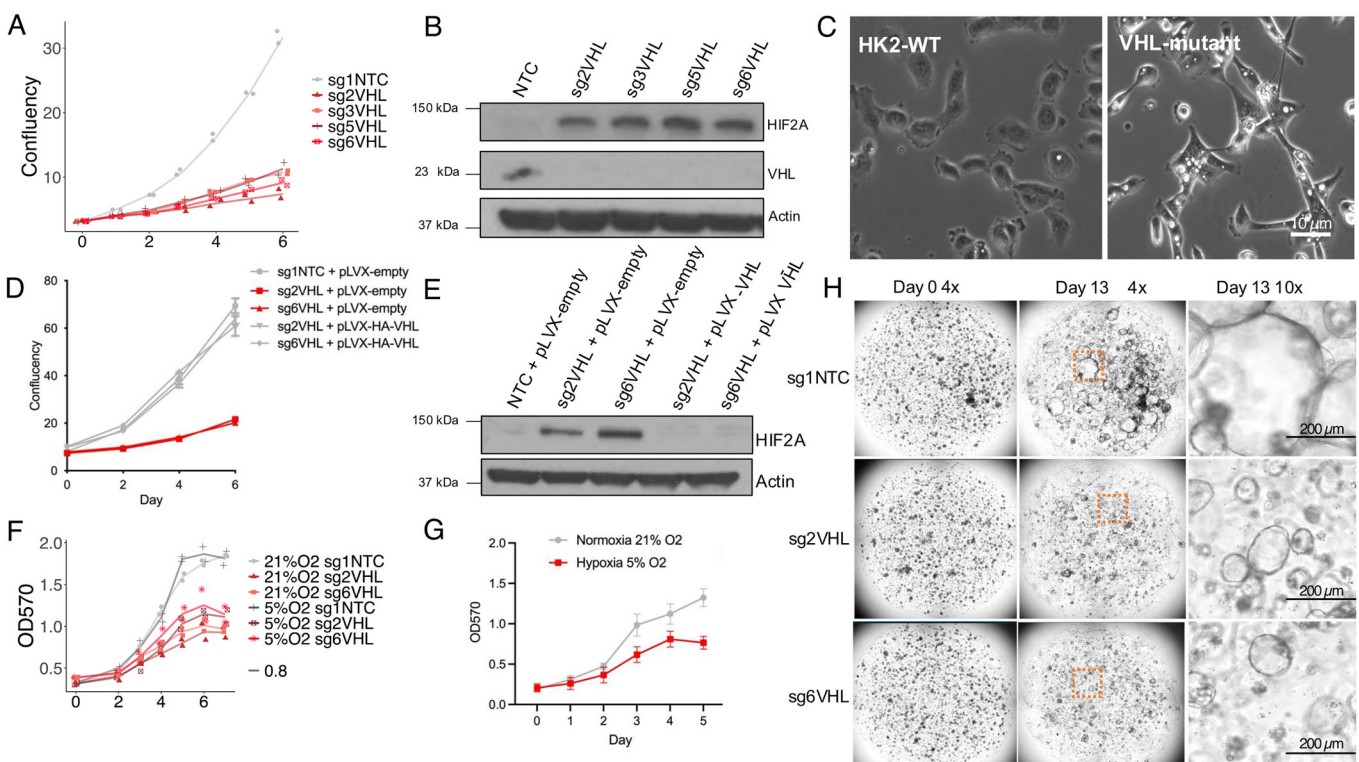

**Figure EV1.  VHL inactivation inhibits cell proliferation.**

(**A**) Proliferation of HK2 cells with or without VHL. $N = 2$ per condition. (mean and S.E.M.). (**B**) Western blot of HIF2A, VHL and Actin on HK2 cells with and without VHL. (**C**) Morphology of HK2 cells with and without VHL. (**D**) Proliferation of VHL mutant HK2 cells with and without VHL re-introduction. $N = 3$ per condition (mean and S.E.M.). (**E**) Western blot of HIF2A on VHL mutant HK2 cells with and without VHL re-introduction. (**F**) Proliferation of HK2 cells with and without VHL under 5% or 21% $O_2$ culture conditions. $N = 2$ per condition (mean and S.E.M.). (**G**) Representative images of human renal epithelial organoids with and without VHL. $N = 4$ replicates per condition. (**H**) Human renal epithelial organoid proliferation under DMSO or DMOG (3 mM) treatment. Source data are available online for this figure.

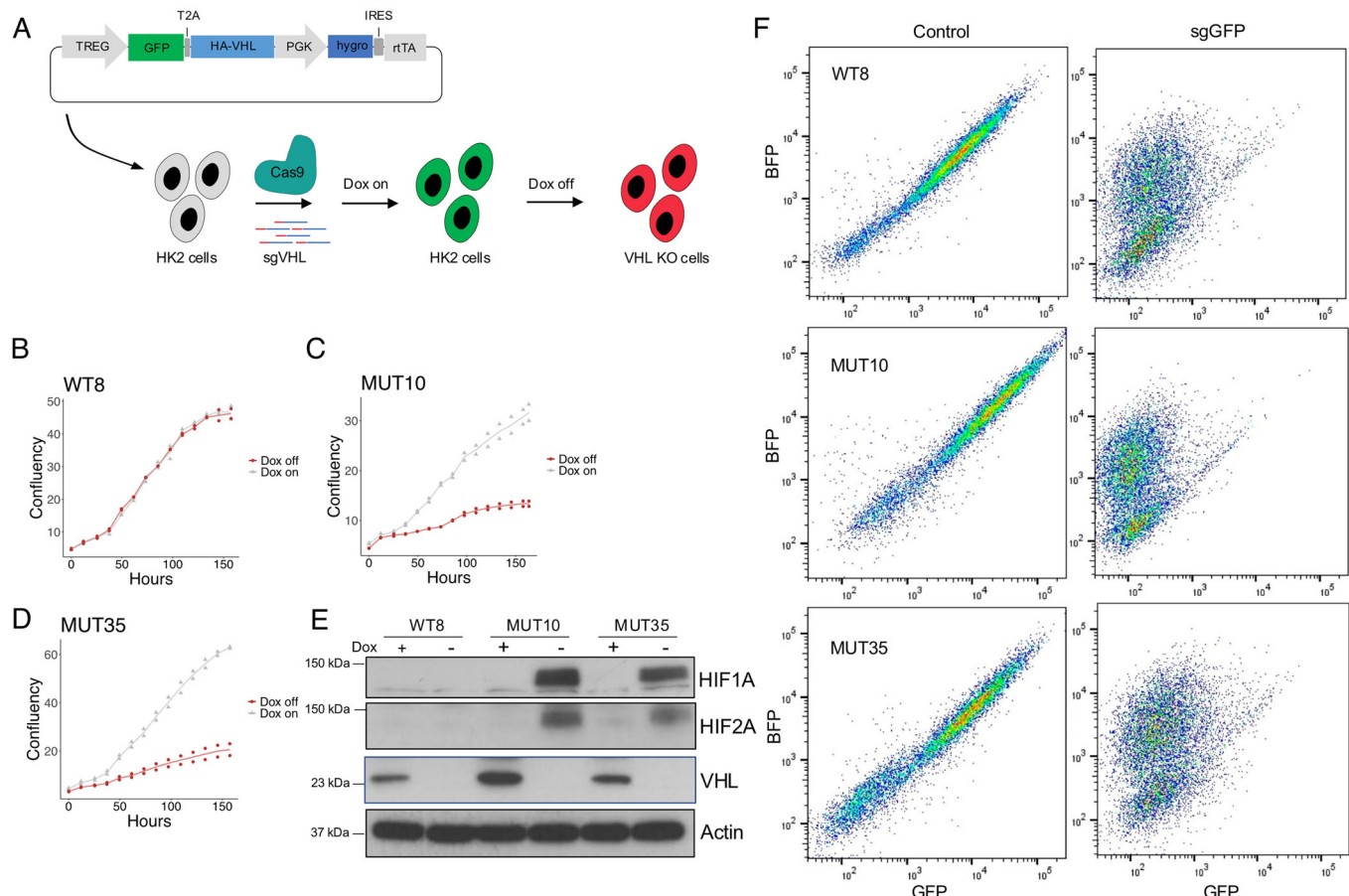

**Figure EV2. Establishment of models with doxycycline-controllable VHL expression.**

(A) Schematic of doxycycline (dox) inducible VHL re-introduction into VHL mutant (MUT10 and MUT35) and wild-type control (WT8) clones. (B–D) Proliferation of WT8 (B), MUT10 (C) and MUT35 (D) cells with and without dox. $N = 2$ replicates per condition (mean and S.E.M.). (E) Western blot of HIF1A, HIF2A, VHL and Actin on WT8, MUT10 and MUT35 cells with and without dox. (F) Cas9 editing efficiency tested on WT8, MUT10 and MUT35 cells by a reporter plasmid using fluorescence-activated cell sorting. Source data are available online for this figure.

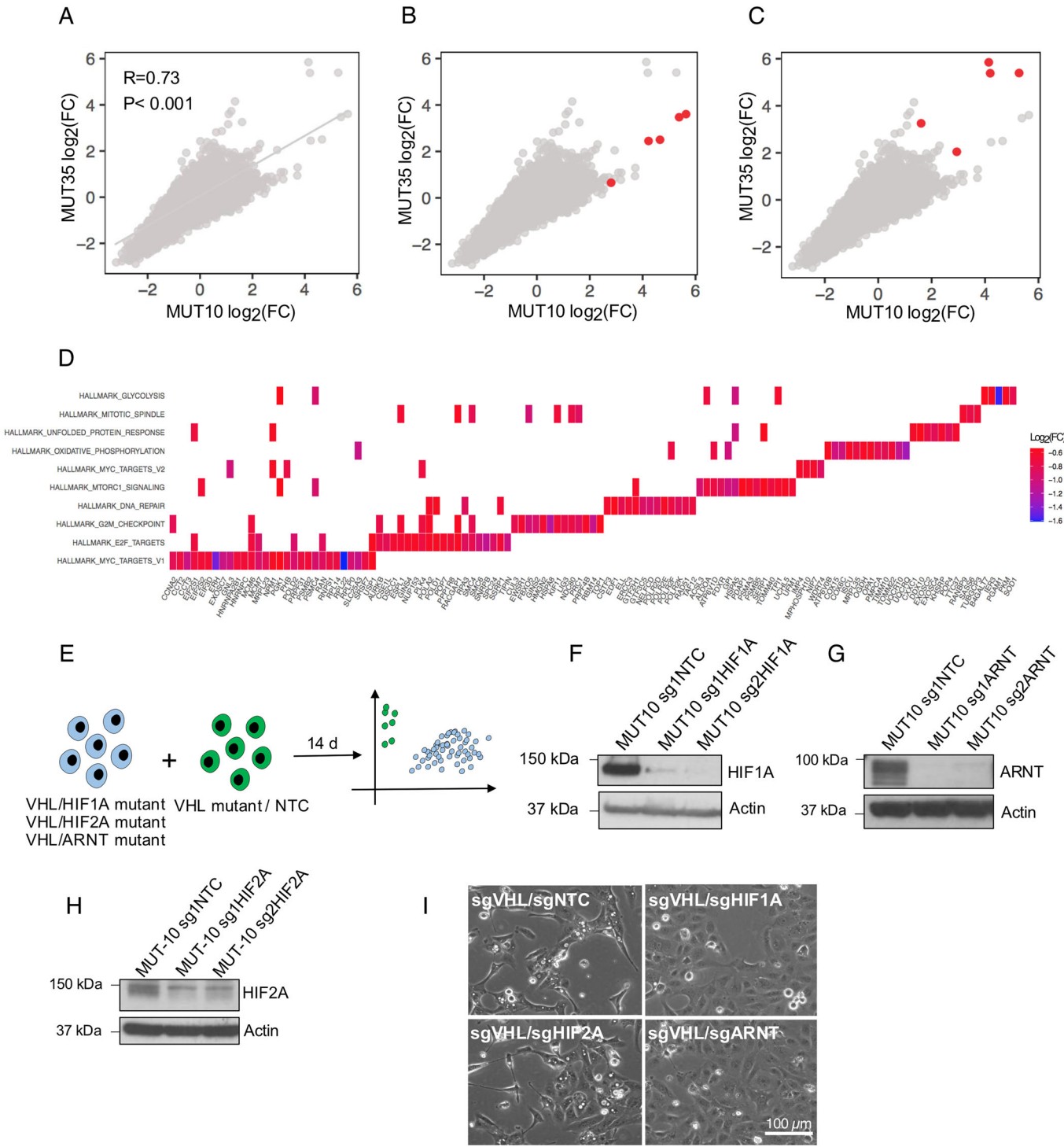

**Figure EV3.   HIF1A and ARNT inhibit proliferation upon VHL inactivation.**

(A) CRISPR/Cas9-based genome wide screen data. sgRNA abundance on day 28 relative to start of the assay in VHL mutant clones MUT10 and MUT35. R, Pearson's correlation coefficient. (B, C) As in (A) with sgRNAs targeting genes of interest highlighted in red: HIF1A in (B) and ARNT in (C). (D) Pathway enrichment analysis on the top 500 genes the sgRNAs of which are depleted over time in MUT10 and MUT35 cells using the Cancer Hallmarks gene sets. (E) A schematic of the competitive proliferation assay. VHL-HIF1A, VHL-HIF2A and VHL-ARNT double mutant cells (BFP labelled) competed against VHL-NTC single mutant cells (GFP labelled). (F–H) Western blot of HIF1A, ARNT and HIF2A on MUT10 cells with and without HIF1A, ARNT or HIF2A inactivation, respectively. (I) Morphology of VHL-NTC, VHL-HIF1A, VHL-HIF2A and VHL-ARNT cells. Source data are available online for this figure.

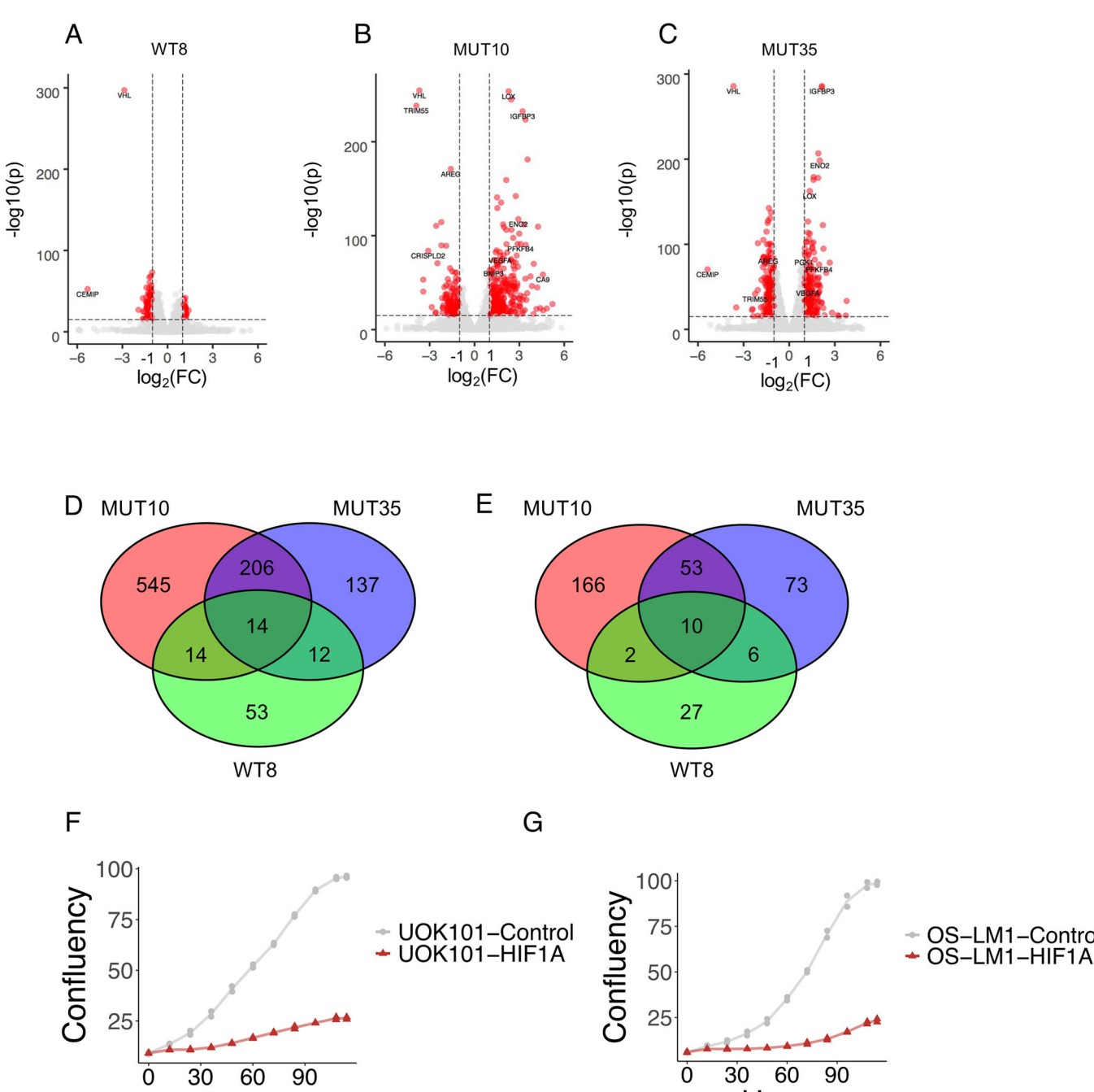

**Figure EV4. HIF1A inhibits mitochondrial function.**

(A–C) Differential gene expression analysis by RNA-seq. WT8, MUT10 and MUT35 cells, dox off compared to dox on. Adjusted *P* value and fold change determined by DESeq2, using the Wald test to test the significance. *N* = 4 replicates per condition. (D) Venn diagram of upregulated genes upon dox withdrawal for WT8, MUT10 and MUT35 cells. (E) Venn diagram of downregulated genes upon dox withdrawal for WT8, MUT10 and MUT35. (F) Proliferation of UOK101 ccRCC cells with and without HIF1A cDNA expression. *N* = 2 replicates per condition (mean and S.E.M.). (G) Proliferation of OS-LM1 ccRCC cells with and without HIF1A cDNA expression. *N* = 2 replicates per condition (mean and S.E.M.). Source data are available online for this figure.

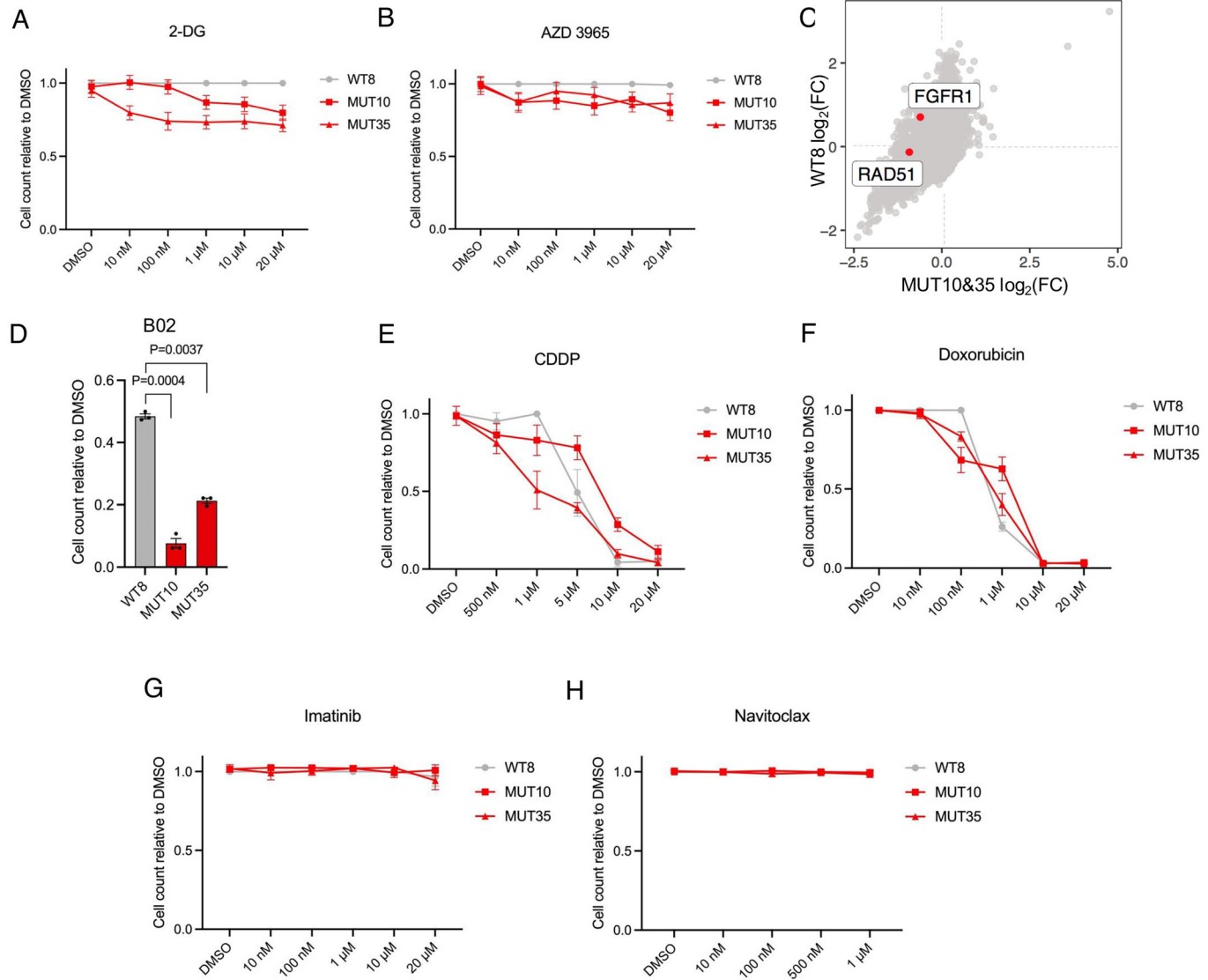

**Figure EV5.** *VHL* **mutant cells show druggable genetic vulnerabilities.**

(A, B) Cell counts on day 5 relative to DMSO control group under 2-DG (10 nM, 100 nM, 1 μM, 10 μM, 20 μM) and AZD3965 (10 nM, 100 nM, 1 μM, 10 μM, 20 μM) treatments. $N = 4$ per condition (mean and SD). (C) CRISPR-Cas9 screen data. FGFR1and RAD51 gene beta score distribution in in VHL WT and mutant cells. (D) Cell counts on day 5 relative to DMSO control group under B02 (10 μM). $N = 3$ replicates per condition (mean and SD). Paired *t* test. (E–H) Cell counts on day 5 relative to DMSO control group under CDDP (10 nM, 100 nM, 1 μM, 10 μM, 20 μM), Doxorubicin (10 nM, 100 nM, 1 μM, 10 μM, 20 μM), Imatinib (10 nM, 100 nM, 1 μM, 10 μM, 20 μM) and Navitoclax (10 nM, 100 nM, 1 μM, 10 μM, 20 μM). $N = 3$ replicates per condition (mean and SD). Source data are available online for this figure.

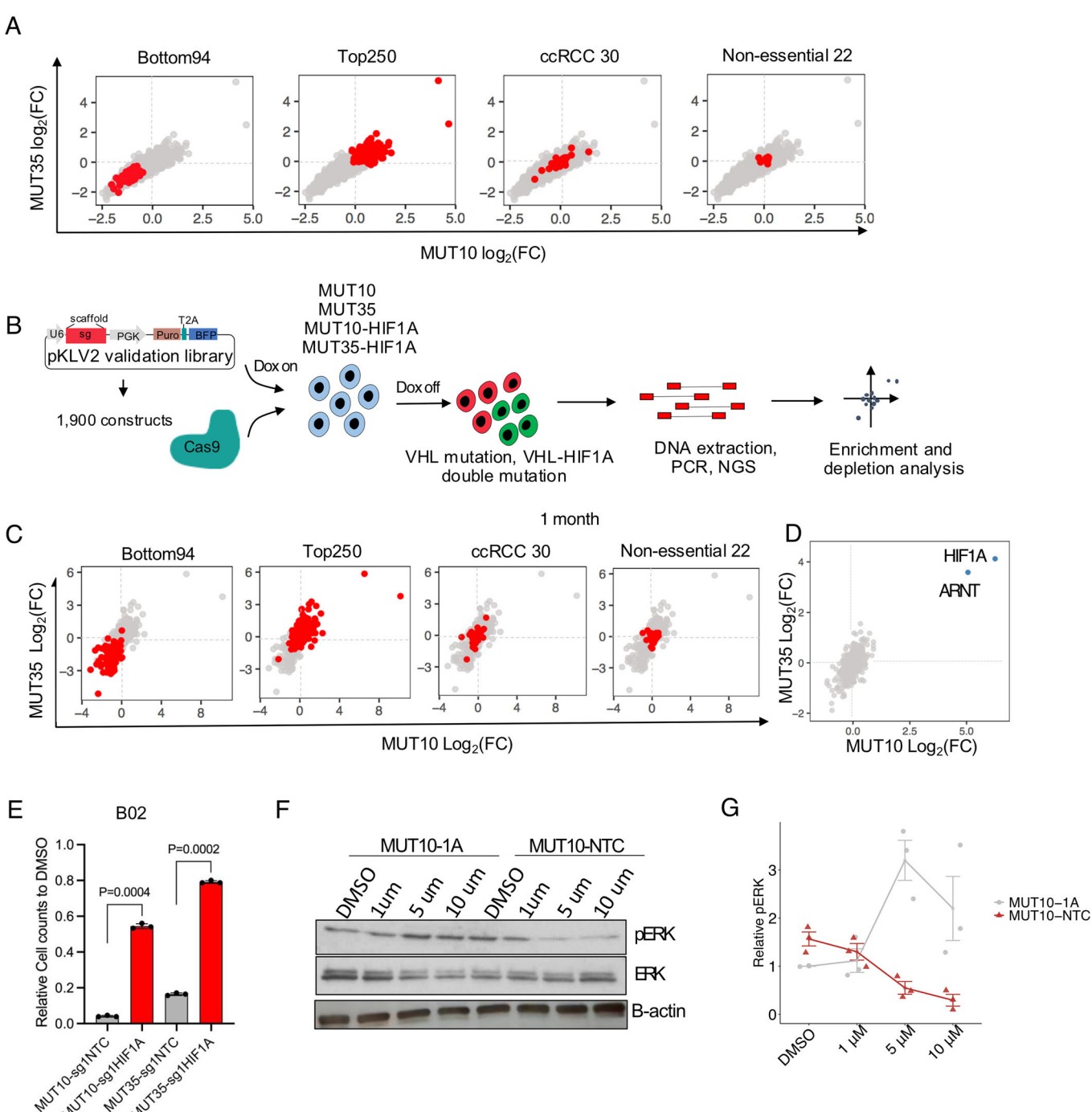

**Figure EV6. CRISPR-Cas9 validation screen.**

(**A**) Genes selected for the validation screen. Distribution of beta scores in the genome-wide CRISPR-Cas9 screen data set. 94 genes the constructs of which were specifically depleted in *VHL* mutant cells, 250 the constructs of which were specifically enriched in *VHL* mutant cells, 30 frequently mutated ccRCC genes and 22 non-essential control genes. (**B**) Schematic of the validation screen on MUT10, MUT35, MUT10-sgHIF1A and MUT35-sgHIF1A cells. (**C**) CRISPR-Cas9-based validation screen data. Gene level construct abundance relative to start of the assay in MUT10 and MUT35 cells. Gene sets of interest highlighted in red. (**D**) CRISPR-Cas9-based validation screen data, doxycycline withdrawal before sgRNA transduction. Gene level construct abundance relative to start of the assay in MUT10 and MUT35 cells. (**E**) Cell counts of B02 (10 μM) treated cells on day 5 relative to DMSO control group. $N = 3$ replicates per condition (mean and SD). Student's *t* test. (**F**) WB of MUT1-1A and MUT10-NTC treated with DMSO, 1 μM, 5 μM, 10 μM Pazopanib for pERK, total ERK and beta-actin. (**G**) Quantification of relative pERK at different treatment concentration at indicated cell population. $N = 3$ replicates per condition (mean and S.E.M.). Source data are available online for this figure.

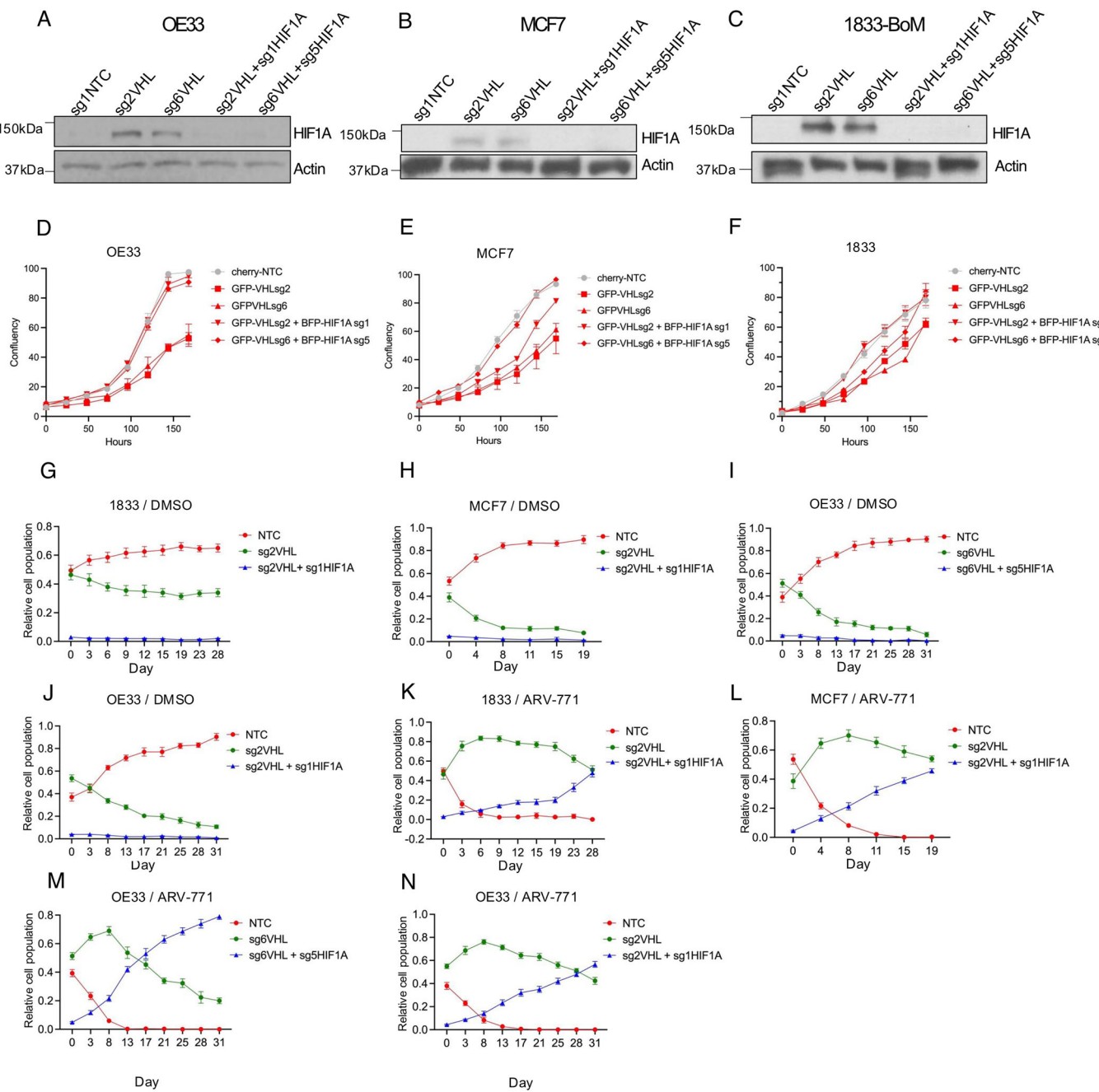

**Figure EV7. HIF1A loss-mediated escape from PROTAC-induced growth inhibition in cancer.**

(A–C) Western blot of HIF1A and Actin on WT (NTC), VHL mutant (VHLsg2, VHLsg6) and VHL-HIF1A double mutant (VHLsg2/HIF1Asg1, VHLsg6/HIF1Asg5) OE33 (A), MCF7 (B) and 1833-BoM (C) cells. (D–F) Incucytes proliferation of OE33, MCF7 and 1833 under different genotype. $N = 3$. (G–N) FACS-based quantification of the relative abundances of different cell populations in competition assays. 1833-BoM cells: 500 nM ARV-771; MCF7 cells: 200 nM ARV-771; OE33 cells: 400 nM ARV-771. $N = 3$ for each condition and timepoint (mean and SD). Source data are available online for this figure.

