## [Peer Review File · EMBO Molecular Medicine]

Mechanisms of resistance to VHL loss-induced genetic and pharmacological vulnerabilities

Jianfeng Ge, Shoko Hirose, Leticia Castillon, Saroor Patel, Ludovic Wesolowski, Anna Dyas, Cissy Yong, Sanne de Haan, Jarno Drost, Grant Stewart, Anna Obenauf, Daniel Munoz-Espin, and Sakari Vanharanta

Corresponding authors: Sakari Vanharanta (sakari.vanharanta@helsinki.fi) , Daniel Munoz-Espin (dm742@cam.ac.uk)

Review Timeline:

Submission Date:	13th Jun 25
Editorial Decision:	9th Jul 25
Revision Received:	3rd Oct 25
Editorial Decision:	28th Oct 25
Revision Received:	11th Nov 25
Editorial Decision:	20th Nov 25
Revision Received:	26th Nov 25
Accepted:	1st Dec 25

Editor: Lise Roth

Transaction Report:

9th Jul 2025

Dear Dr. Vanharanta,

Thank you for submitting your new manuscript to EMBO Molecular Medicine.

We have now received feedback from the three reviewers who had already reviewed the previous version of your manuscript. As you will see from the reports below, the referees acknowledge the revisions that were performed, and are overall supporting publication of your work pending appropriate revisions.

In order for us to consider the manuscript further, it is necessary for you to address the remaining reviewers' concerns in full, and acceptance of the manuscript will entail another round of review. Particular attention should be given to ensuring the clarity of the text and figures.

We are expecting your revised manuscript within three months, if you anticipate any delay, please contact us.

We require:

4) A .docx formatted letter INCLUDING the reviewers' reports and your detailed point-by-point responses to their comments. As part of the EMBO Press transparent editorial process, the point-by-point response is part of the Review Process File (RPF), which will be published alongside your paper.

5) A complete author checklist, which you can download from our author guidelines (<https://www.embopress.org/page/journal/17574684/authorguide#submissionofrevisions>). Please insert information in the checklist that is also reflected in the manuscript. The completed author checklist will also be part of the RPF.

6) All Materials and Methods need to be described in the main text using our 'Structured Methods' format. According to this format, the Methods section includes a Reagents and Tools Table (listing key reagents, experimental models, software and relevant equipment and including their sources and relevant identifiers) followed by a Methods and Protocols section describing the methods, ideally using a step-by-step protocol format. The aim is to facilitate adoption of the methodologies across labs. Please download and fill our Reagents and Tools Table template (.docx), which you can find in our author guidelines: <https://www.embopress.org/page/journal/14693178/authorguide#structuredmethods>.

7) Please note that all corresponding authors are required to supply an ORCID ID for their name upon submission of a revised manuscript.

8) It is mandatory to include a 'Data Availability' section after the Materials and Methods. Before submitting your revision, primary datasets produced in this study need to be deposited in an appropriate public database, and the accession numbers and database listed under 'Data Availability'. Please remember to provide a reviewer password if the datasets are not yet public (see <https://www.embopress.org/page/journal/17574684/authorguide#dataavailability>).

9) For data quantification: please specify the name of the statistical test used to generate error bars and P values, the number

(n) of independent experiments (specify technical or biological replicates) underlying each data point and the test used to calculate p-values in each figure legend. The figure legends should contain a basic description of n, P and the test applied. Graphs must include a description of the bars and the error bars (s.d., s.e.m.). Please provide exact p values.

10) Our journal encourages inclusion of *data citations in the reference list* to directly cite datasets that were re-used and obtained from public databases. Data citations in the article text are distinct from normal bibliographical citations and should directly link to the database records from which the data can be accessed. In the main text, data citations are formatted as follows: "Data ref: Smith et al, 2001" or "Data ref: NCBI Sequence Read Archive PRJNA342805, 2017". In the Reference list, data citations must be labeled with "[DATASET]". A data reference must provide the database name, accession number/identifiers and a resolvable link to the landing page from which the data can be accessed at the end of the reference. Further instructions are available at .

11) We replaced Supplementary Information with Expanded View (EV) Figures and Tables that are collapsible/expandable online. EV Figures should be cited as 'Figure EV1, Figure EV2' etc... in the text and their respective legends should be included in the main text after the legends of regular figures.

12) The paper explained: EMBO Molecular Medicine articles are accompanied by a summary of the articles to emphasize the major findings in the paper and their medical implications for the non-specialist reader. Please provide a draft summary of your article highlighting

13) Author contributions: CRedit has replaced the traditional author contributions section because it offers a systematic machine readable author contributions format that allows for more effective research assessment. Please remove the Authors Contributions from the manuscript and use the free text boxes beneath each contributing author's name in our system to add specific details on the author's contribution. More information is available in our guide to authors.

Please also suggest a visual abstract to illustrate your article as a PNG file 550 px wide x 300-600 px high. A cropped portion of this image will serve as thumbnail for the table of content on our webpage.

16) As part of the EMBO Publications transparent editorial process initiative (see our Editorial at <http://embomolmed.embopress.org/content/2/9/329>), EMBO Molecular Medicine will publish online a Review Process File (RPF) to accompany accepted manuscripts.

In the event of acceptance, this file will be published in conjunction with your paper and will include the anonymous referee reports, your point-by-point response and all pertinent correspondence relating to the manuscript. Let us know whether you agree with the publication of the RPF and as here, if you want to remove or not any figures from it prior to publication. Please note that the Authors checklist will be published at the end of the RPF.

I look forward to receiving your revised manuscript.

Yours sincerely,

Lise Roth

***** Reviewer's comments *****

Referee #1 (Remarks for Author):

Is suitable for publication

Referee #2 (Comments on Novelty/Model System for Author):

The technical quality, including the use of screens, sgRNAs, and rescue experiments is strong. Concerns around the novelty and impact of the work has been addressed based on the comments in the first round of reviews, including new in vivo data and the use of VHL mutants.

Referee #2 (Remarks for Author):

The authors have addressed many of the concerns raised in the first round of reviews and also added new experimental data - both in vivo and in vitro. While these studies have strengthened the manuscript, there are still some concerns that need to be addressed. Although, some of these concerns can be addressed with better data presentation and textual edits, the lack of the empty vector control in Fig. 5 is a notable oversight and will need to be addressed with new experimental data.

Specific comments:

1. In Fig. 4 and S7, the behavior of the blue cell population (sgVHL+sgHIF1a) is very confusing. If these cells are able to overcome the anti-proliferative effects of VHL loss (as indicated in S7D-S7F), why do these cell populations show no enrichment even over the course of 2-3 weeks in the DMSO arm?
2. The colors have been switched in Fig. 5D (sgNT cells are now blue and sgVHL+sgHIF1 cells are labeled red). Nevertheless, the results here are also a bit confusing. For example, why do the control arms not have greater representation of BFP and mCherry cells? These cells should be increasing in representation at the cost of the sgVHL (green) cells.
3. While it is appreciable that the authors have addressed the role of disease-associated VHL mutants in their studies and also seen some interesting results, there are some concerns with this experiment.

First, the results of this experiment cannot be interpreted without an empty vector control. Second, the VHL type 2 mutants are sub-classified, based on disease associations, into 2A, 2B, or 2C. These mutants have entirely different impacts on HIF proteolysis. The 2B mutants, such as Y98N, cannot degrade HIF [Li L et al., 2007, Mol Cell Biol. (15):5381-92.], whereas the type 2A and 2C mutants (e.g., Y98H and L188V, respectively) retain the ability to target HIF. In this regard, the choice to generate the Y98F mutant is unclear, considering that it is not a common disease-associated variant of VHL. Perhaps, the Y98F mutants pheno-copy the Y98H mutation and thus rescue the proliferation defect akin to WT VHL. However, these functional differences are not evident from the HIF blot in 5E, which shows relatively modest differences in HIF abundance (another reason why an empty vector control is necessary). Similarly, the choice of the P99 mutants, which are also not commonly studied disease variants of VHL, is conceptually unclear.

4. Lines 148-151: The statement, "However, combined analysis of MUT10 and MUT35 cells revealed 220 genes significantly upregulated and 63 genes significantly downregulated upon dox withdrawal and consequent VHL depletion (Fig. 2A, Fig. S4B-C and Supplementary Table 2-3), could be presented as a Venn diagram +/- Dox for the three cell lines.

5. 2E, G, I, K: labels say OCR, but legend says ATP production. Please clarify what these graphs are plotting.

6. Line 245: "translateed" typo

7. Line 252: "incomparision" typo

Referee #3 (Remarks for Author):

This is my 2nd time to review this manuscript. I did notice the authors added a new Figure 5. However,

1. Almost no revision was demonstrated in both figures and description to address the comments from the previous reviewers.
2. This manuscript is lack of novelty. Vhl loss reduced Renca tumor growth has been reported by different groups, and HIF1A-dependent mitochondrial inhibition is also known.
3. This manuscript is lack of focus. From FGFR1 to RAD51 to PROTAC therapy, it is hard to follow. None of their relationship to HIF1A was comprehensively investigated.
4. This manuscript is lack of clinical application. Since VHL loss resulted in HIF1a-dependent proliferative inhibition, why do we still seek to target these cells?
5. The PROTAC therapy section is also confusing, probably due to my limited background. VHL-mediated HIF1A degradation does not require ARV-771. Why is it important to use this drug? To promote the degradation of bromodomain and extra-terminal domain (BET) proteins? What are they in the current system? What type of tumors and neoplastic tissues harbors double mutant cells? Why is it important to restore the proliferative fitness of double mutant cells?
6. Maybe only for me, the writing is not concise and hard to understand.

Revised version of Ge et al. "Mechanisms of resistance to VHL loss-induced genetic and pharmacological vulnerabilities"**Point-by-point response to referee comments**

We would like to thank the referees for taking the time to review our work. We really appreciate the constructive comments that have allowed us to improve our study. Below we address each point in detail. The most significant change in this revised version is the addition of new mutant *VHL* constructs into the experiments shown in **Fig. 5E-G**. We have also added explanatory text in the results section to improve readability, replotted Fig. 5D to improve the presentation and updated the data set for Fig. 1A. We have also redrawn the schematics in Fig. 5 to improve clarity. With the submission we provide a copy of the revised text in which changes are highlighted.

Referee #1:

"Is suitable for publication"

Reply: We thank the referee for taking the time to review our work.

Referee #2:

"The technical quality, including the use of screens, sgRNAs, and rescue experiments is strong. Concerns around the novelty and impact of the work has been addressed based on the comments in the first round of reviews, including new in vivo data and the use of VHL mutants."

Reply: We thank the referee for the positive comments and taking the time to review our work.

"The authors have addressed many of the concerns raised in the first round of reviews and also added new experimental data - both in vivo and in vitro. While these studies have strengthened the manuscript, there are still some concerns that need to be addressed. Although, some of these concerns can be addressed with better data presentation and textual edits, the lack of the empty vector control in Fig. 5 is a notable oversight and will need to be addressed with new experimental data."

Reply: We thank the referee for the positive comments and taking the time to review our work. As discussed in detail below, we have now thoroughly addressed all the remaining points with new data and by editing the text.

"Specific comments:

1. In Fig. 4 and S7, the behavior of the blue cell population (sgVHL+sgHIF1a) is very confusing. If these cells are able to overcome the anti-proliferative effects of VHL loss (as indicated in S7D-S7F), why do these cell populations show no enrichment even over the course of 2-3 weeks in the DMSO arm?"

Reply: Thank you for the comment, this is an important point. We believe the reason is that all three cell populations are competing against each other. As these experiments are done with cell populations, we do not expect that in the double mutant blue population 100% of cells have the intended mutations. Thus, at the population level, it is expected that the red control population remains fitter and thus prevents the blue population from becoming more abundant in the absence of drug treatment, even if the green population gradually disappears. However,

this dynamic is dramatically altered upon ARV-771 treatment (**Fig. 4G-H**), showing that the blue population is competent to proliferate.

“2. The colors have been switched in Fig. 5D (sgNT cells are now blue and sgVHL+sgHIF1 cells are labeled red). Nevertheless, the results here are also a bit confusing. For example, why do the control arms not have greater representation of BFP and mCherry cells? These cells should be increasing in representation at the cost of the sgVHL (green) cells.”

Reply: Thank you for the comment. We would like to clarify that **Fig. 5D** shows the relative abundance of the different cell populations in control and ARV-771-treated tumors. These data are based on flow cytometry. We have normalized the data to the control tumours, an approach we think is the best for this experiment as, unlike in the in vitro experiments, it is not possible to sample the cell population continuously. In the normalized presentation, the control arms have a relative representation of 1 for each cell type, while in the ARV-771-treated tumours the control cells have become less abundant, double *VHL* / *HIF1A* mutant cells have become more abundant, and single *VHL* mutant cells have stayed unchanged. These data are consistent with our in vitro data, thus providing in vivo validation to our model. We have amended the text and data panel to clarify the presentation.

“3. While it is appreciable that the authors have addressed the role of disease-associated VHL mutants in their studies and also seen some interesting results, there are some concerns with this experiment.

First, the results of this experiment cannot be interpreted without an empty vector control. Second, the VHL type 2 mutants are sub-classified, based on disease associations, into 2A, 2B, or 2C. These mutants have entirely different impacts on HIF proteolysis. The 2B mutants, such as Y98N, cannot degrade HIF [Li L et al., 2007, Mol Cell Biol. (15):5381-92.], whereas the type 2A and 2C mutants (e.g., Y98H and L188V, respectively) retain the ability to target HIF. In this regard, the choice to generate the Y98F mutant is unclear, considering that it is not a common disease-associated variant of VHL. Perhaps, the Y98F mutants pheno-copy the Y98H mutation and thus rescue the proliferation defect akin to WT VHL. However, these functional differences are not evident from the HIF blot in 5E, which shows relatively modest differences in HIF abundance (another reason why an empty vector control is necessary). Similarly, the choice of the P99 mutants, which are also not commonly studied disease variants of VHL, is conceptually unclear.”

Reply: Thank you for the comment, we completely agree that the empty vector condition is essential for these experiments. This has now been corrected and the experiment has been repeated. Also, as suggested, we have added two new mutations, Y98N and Y98H, and optimized the Western blot. While the different phenotypes of type 2A-C *VHL* mutations suggested the possibility that PROTAC- and HIF1A-related functions of VHL could be dissociated, we based our original choice of mutations in residues Y98 and P99 on the recent functional data from a saturation mutagenesis screen, which indicated that mutations in residues Y98 and P99 may give resistance to ARV-771 (1). We find that Y98N, Y98F, Y98H and P99G provide resistance to ARV-771, whereas P99M does not. In addition, we find that Y98N, Y98H and Y98G do not degrade HIF1A, but Y98F and P99M do. These data are in line with the proliferation data, which shows that cells expressing *VHL* mutants that can target HIF1A proliferate faster. We note that Y98H does not degrade HIF1A in our experiment, even though it is classified as a type 2A mutant and shows HIF2A degradation in the paper by Li et al. (2). Examination of the data by Li et al. shows, however, that the effect of Y98H on HIF1A is weaker than what is seen for HIF2A in RCC4 cells, and that wildtype VHL controls HIF1A expression better than Y98H. Based on this, our data are consistent with the data by Li et al., especially since the expression of Y98H in our experiment appears weaker than the expression of the other mutants, possibly because of technical reasons. The new data are

presented in the revised panels **Fig. 5E-G**. We have also amended the associated text to clarify how the mutations were selected for this experiment.

“4. Lines 148-151: The statement, “However, combined analysis of MUT10 and MUT35 cells revealed 220 genes significantly upregulated and 63 genes significantly downregulated upon dox withdrawal and consequent VHL depletion (Fig. 2A, Fig. S4B-C and Supplementary Table 2-3), could be presented as a Venn diagram +/- Dox for the three cell lines.”

Reply: Thank you for the suggestion, we have generated Venn diagrams and include them as a new panels **D and E** in **EV4**.

“5. 2E, G, I, K: labels say OCR, but legend says ATP production. Please clarify what these graphs are plotting.”

Reply: Thank you for highlighting this. We have revised the figure legend to clarify this point.

“6. Line 245: “translateed” typo”

Reply: Corrected, thank you for highlighting this.

“7. Line 252: “incomparision” typo”

Reply: Corrected, thank you for highlighting this.

Referee #3:

*“This is my 2nd time to review this manuscript. I did notice the authors added a new Figure 5. However,
1. Almost no revision was demonstrated in both figures and description to address the comments from the previous reviewers.”*

Reply: We thank the referee for taking the time to review our work, we really appreciate the constructive comments from this and the previous round. They have allowed us to significantly improve our study. Since the first submission, we have conducted a significant body of new experiments, leading to substantial new data sets that together with extensive rewriting of the text have addressed all the referee points. For clarification, our study contains the following components:

- using unbiased genome wide CRISPR-Cas9-based functional screening in novel experimental systems, with validation in human primary cells, we demonstrate that HIF1A/ARNT is the critical mediator of VHL loss-induced fitness loss in human cells
- by comparing *VHL* null and wildtype cells, we identify specific genetic vulnerabilities associated with VHL inactivation
- through a secondary genetic screen, we demonstrate that VHL loss-induced genetic vulnerabilities are caused by HIF1A activation
- we identify pharmacologically targetable genetic vulnerabilities in *VHL* null cells and demonstrate that FGFR1 and RAD51 inhibitors have a specific growth suppressive effect on *VHL* mutant cells, with new data on specificity and the biochemical read-outs of FGFR1 inhibitors included after the first round of revisions

- we demonstrate that combined VHL / HIF1A inactivation in breast and esophageal cancer cells causes resistance to ARV-771, a member of the emerging class of VHL-based PROTACs that has shown anti-cancer activity in several tumor contexts
- demonstration of the selective advantage of VHL / HIF1A double mutant cells under ARV-771 treatment in vivo, a significant data set presented in the revised **Fig. 5**
- as a new conceptual advancement, included after the first round of revision and now further explored with new mutations and improved data sets (**Fig. 5E-H**), the identification of *VHL* mutations that dissociate the roles VHL has in mediating PROTAC effects and HIF1A degradation, indicating that specific *VHL* mutations could provide PROTAC resistance without simultaneously causing reduced proliferative fitness induced by HIF1A

Collectively, we present a substantial body of work that has been extensively expanded with conceptually novel results and important clarifications during the revision process. In this second revised version of the manuscript we have further edited the text to clarify the central points of our study. As explained in detail in our next response below, we hope that these changes clarify the contribution of our study to the field.

“2. This manuscript is lack of novelty. Vhl loss reduced Renca tumor growth has been reported by different groups, and HIF1A-dependent mitochondrial inhibition is also known.”

Reply: Even though our work has confirmed and reinforced some previous observations, giving validity to our study, it contains several aspects that are new. For example, while we agree that the link between VHL and HIF1A has been known, there has been significant controversy about the mechanisms through which *VHL* loss may lead to reduced cellular fitness, with prominent papers suggesting that this may happen in a HIF1A independent manner (3–5). Moreover, even though genetic screens for synthetic lethal *VHL* interactions have been recently published (6,7) these studies have been conducted using *VHL* null renal cancer cell lines, which by definition have circumvented the mechanisms that lead to reduced cellular fitness after *VHL* loss. Thus, our approach that uses genetic engineering in *VHL* wildtype cells and large unbiased genetic screening is, to our knowledge, the first attempt to systematically map the pathways that induce negative fitness effects upon VHL loss in non-cancerous cells, identifying a single but key central mediator: the HIF1A/ARNT complex. Importantly, our data provide a proof of principle that these effects can be specifically targeted using pharmacological agents, a critical first step towards exploiting the effects of HIF1A/ARNT activation for ccRCC prevention in high-risk patients, such as *VHL* mutation carriers. These results also suggest that targeting the inappropriate activation of HIF1A/ARNT should be the central approach in the efforts to help patients suffering from *VHL* loss-induced systemic metabolic dysfunction (8). Finally, our demonstration of PROTAC insensitivity in *VHL* / *HIF1A* mutant cells across cancer types, validated in an in vivo tumor model, suggests approaches towards reducing the likelihood of therapy resistance for this new but prominent class of emerging cancer therapeutics. Our results therefore go far beyond the current literature and provide a valuable addition to knowledge and potentially translatable tools and concepts for the development of novel therapeutic strategies. We have edited the revised manuscript to further clarify these points.

“3. This manuscript is lack of focus. From FGFR1 to RAD51 to PROTAC therapy, it is hard to follow. None of their relationship to HIF1A was comprehensively investigated.”

Reply: The underlying rationale of our work stems from three contexts where *VHL* loss plays or may play a clinically relevant role: renal cancer, biallelic *VHL* deficiency and VHL-dependent PROTACs, which are not yet in routine clinical use but are explored as possible therapeutic agents for human disease. Our interpretation of the data in **Fig. 3** and **Fig. 4-5** is that HIF1A activation upon VHL loss can be a vulnerability in different contexts: it can sensitize cells to pharmacological agents such as the FGFR1 inhibitor ACD4547 in a HIF1A-dependent manner (**Fig. 3**), and it can prevent the proliferation and clonal dominance of *VHL* null cells upon

PROTAC treatment (**Fig. 4-5**). Importantly, the revised version of our study also identifies *VHL* mutations that dissociate the effects *VHL* has on HIF1A stabilization and PROTAC activity (**Fig. 5E-G**). Thus, our study thus has a very clear focus: to study the effects and consequences of *VHL* loss in clinically relevant contexts that have thus far not been explored in functional experiments. Also, as described in detail in the points 1 and 2 above, our study contains substantial new data sets that go far beyond the current literature. However, we agree with the referee that open questions remain. We have highlighted this in the revised discussion of the manuscript.

“4. This manuscript is lack of clinical application. Since VHL loss resulted in HIF1a-dependent proliferative inhibition, why do we still seek to target these cells?”

Reply: We agree that potential clinical applicability is important. Indeed, as described in our manuscript, *VHL* loss is clinically relevant in at least three context. First, *VHL* loss leads to renal and other tumors in individuals that carry heterozygous *VHL* mutations in the germline. Also, biallelic *VHL* loss is a common cause of sporadic renal cancer. However, in both the familial and sporadic cases, *VHL* loss, while eventually leading to tumors, does not cause tumorigenesis in all susceptible cells, and even when it does, the process takes a long time, up to several decades, providing an extensive temporal window of opportunity for early intervention before aggressive cancer clones emerge. However, our poor understanding of the molecular vulnerabilities of these pre-cancerous *VHL* null clones has hampered the development rational intervention strategies. Our new results highlight the role of HIF1A-induced vulnerabilities in acutely *VHL* null cells, providing a proof of concept that these could be therapeutically targeted, with the potential consequence being that cancer development could be reduced, especially in individuals carrying germline *VHL* mutations. Second, biallelic germline mutations in *VHL* can cause a severe systemic metabolic defect (8). Our results suggest that HIF1A activation may be the central mediator of this phenotype, highlighting the relevance of developing strategies for selective HIF1A inhibition. Third, *VHL* inactivation can make cells insensitive to *VHL*-dependent PROTACs, and our findings provide insight into the mechanisms that can allow cancer cells to evade the fitness defect caused by *VHL* loss, thus making them functionally resistant to PROTACs. In conclusion, *VHL* null cells are clinically relevant in several different contexts and understanding their biology is important for the development of new therapeutic approaches. This is an important point that we have clarified further in the revised manuscript.

“5. The PROTAC therapy section is also confusing, probably due to my limited background. VHL-mediated HIF1A degradation does not require ARV-771. Why is it important to use this drug? To promote the degradation of bromodomain and extra-terminal domain (BET) proteins? What are they in the current system? What type of tumors and neoplastic tissues harbors double mutant cells? Why is it important to restore the proliferative fitness of double mutant cells?”

Reply: The referee is correct, HIF1A degradation by *VHL* does not require ARV-771. However, this does not mean that ARV-771 or other *VHL*-dependent PROTACs are uninteresting as a drugs. On the contrary, given the reduced fitness caused by *VHL* loss in multiple cell types, *VHL*-dependent PROTACs have significant potential as cancer therapeutics, as reviewed here (9). ARV-771 targets bromodomain proteins, most notably BRD2, 3 and 4, as shown previously (10), but *VHL*-dependent PROTACs can target other relevant cancer targets as well, such as SMARCA2/4 (11). Due to their potentially broad applicability in different contexts, it is important to understand the mechanisms that may lead to clinically relevant, proliferation proficient resistance to this class of agents. Previous literature has led to contradicting results regarding the relevance of HIF1A for *VHL* loss-induced proliferative defects (3,4,12). Our results from multiple human cell types now clearly demonstrate that HIF1A-stabilization is a central cause of *VHL* loss-induced loss of proliferative fitness in many cell lineages, and importantly, that some *VHL* mutations do not activate HIF1A, yet lead to ARV-771 resistance (new data in **Fig.**

5E-G). Thus, by using genetic analyses and studying HIF1A loss in the context of VHL-dependent PROTAC activity we have been able to reveal important biology related to this class of promising anti-cancer agents. Double *VHL/HIF1A* mutant cells have been described in renal carcinoma (13,14), a tumor type in which HIF1A expression is often lost, even though HIF2A remains expressed. However, the goal of our work is not to restore the proliferative fitness of *VHL/HIF1A* mutant cells. Indeed, our data suggest that such cells already have regained their proliferative capacity. Rather, our aim is to understand the functional consequences of VHL loss. We have amended the text in the revised manuscript to further clarify the underlying logic of our work.

“6. *Maybe only for me, the writing is not concise and hard to understand.*”

Reply: We agree that clear and concise writing is important. We have carefully revised the text to further improve readability and to communicate our results and the concepts of our manuscript in the best possible way.

1. Hanzl A. et al. Functional E3 ligase hotspots and resistance mechanisms to small-molecule degraders. *Nat Chem Biol.* **19**: 323–33, 2023
2. Li L. et al. Hypoxia-inducible factor linked to differential kidney cancer risk seen with type 2A and type 2B VHL mutations. *Mol Cell Biol.* **27**: 5381–92, 2007
3. Young A.P. et al. VHL loss actuates a HIF-independent senescence programme mediated by Rb and p400. *Nat Cell Biol.* **10**: 361–9, 2008
4. Welford S.M. et al. Renal oxygenation suppresses VHL loss-induced senescence that is caused by increased sensitivity to oxidative stress. *Mol Cell Biol.* **30**: 4595–603, 2010
5. Abu-Remaileh M. et al. Total loss of VHL gene function impairs neuroendocrine cancer cell fitness due to excessive HIF2 α activity. *Proc Natl Acad Sci U S A.* **121**: e2410356121, 2024
6. Bertlin J.A.C. et al. VHL synthetic lethality screens uncover CBF- β as a negative regulator of STING. *bioRxiv.* , 2024
7. Sun N. et al. VHL Synthetic Lethality Signatures Uncovered by Genotype-Specific CRISPR-Cas9 Screens. *CRISPR J.* **2**: 230–45, 2019
8. Perrotta S. et al. Effects of Germline VHL Deficiency on Growth, Metabolism, and Mitochondria. *N Engl J Med.* **382**: 835–44, 2020
9. Békés M. et al. PROTAC targeted protein degraders: the past is prologue. *Nat Rev Drug Discov.* **21**: 181–200, 2022
10. Raina K. et al. PROTAC-induced BET protein degradation as a therapy for castration-resistant prostate cancer. *Proc Natl Acad Sci U S A.* **113**: 7124–9, 2016
11. Xiao L. et al. Targeting SWI/SNF ATPases in enhancer-addicted prostate cancer. *Nature.* **601**: 434–9, 2022
12. Hoefflin R. et al. HIF-1 α and HIF-2 α differently regulate tumour development and inflammation of clear cell renal cell carcinoma in mice. *Nat Commun.* **11**: 4111, 2020
13. Culliford R. et al. Whole genome sequencing refines stratification and therapy of patients with clear cell renal cell carcinoma. *Nat Commun.* **15**: 5935, 2024
14. Shen C. et al. Genetic and functional studies implicate HIF1 α as a 14q kidney cancer suppressor gene. *Cancer Discov.* **1**: 222–35, 2011

28th Oct 2025

Dear Dr. Vanharanta,

Thank you for submitting your revised study. We have now received the reports from referees #2 and #3. As you will see below, they are satisfied with the revisions, and I will therefore be able to accept your manuscript once the following editorial concerns are addressed:

1/ Please address the comment from referee #3 on the different sections in the rebuttal letter and/or in the manuscript.

2/ Manuscript text:

- Please indicate in track changes mode any new modification in the text.
- An email bounced for S.deHaan-12@prinsesmaximacentrum.nl, please check and correct if needed.
- Please provide up to 5 keywords.
- Summary should be renamed Abstract.
- In the methods:
 - o Cells: please remove "and they are available from the corresponding author upon reasonable request."
 - o Western blots: please indicate whether membranes were stripped.
 - o Organoids: please provide a statement confirming that the experiments conformed to the principles set out in the WMA Declaration of Helsinki and the Department of Health and Human Services Belmont Report.
 - o Animals: please provide a statement on housing and husbandry conditions.
- Data availability section: please remove "Other datasets and tools generated during the current study are available from the corresponding author on reasonable request."; provide URLs for deposited datasets.
- Please rename 'Declaration of Interests' to 'Disclosure statement and competing interests'.

3/ Figures:

- Where possible, please provide individual data points in the graphs.
- Please correct the title to "Dataset EV1" in the legends and in the citations.
- Please address the queries from our data editors in the figure legends:
 1. Please indicate the statistical test used for data analysis in the legends of figures 1A, 2A, EV4 A-C
 2. Please note that the box plots need to be defined in terms of minima, maxima, centre, bounds of box and whiskers, and percentile in the legend of figure 1A
 3. Please note that information related to n is missing in the legends of figures 1A, 2A, 5F, EV1 G, EV4 A-C; EV5 E-H; EV6 G
 4. Please note that the error bars are not defined in the legends of figures EV5 A-H; EV6 G

4/ Thank you for providing Source Data. Please check the raw data provided for Fig. 3B (WT8) as the two bottom lines are identical. Please also check the source data for Fig. 3F.

5/ Checklist:

Experimental animals/animal observed in or captured from the field: please remove the text in the right column.

6/ Please provide "The paper explained" in the manuscript text file:

EMBO Molecular Medicine articles are accompanied by a summary of the articles to emphasize the major findings in the paper and their medical implications for the non-specialist reader. Please provide a draft summary of your article highlighting

7/ Thank you for providing a nice visual abstract. I have cropped a small portion to serve as thumbnail on our webpage (attached). Please let us know if you agree or provide an alternative image at the right dimensions (115x70 px).

8/ As part of the EMBO Publications transparent editorial process initiative (see our Editorial at

<http://embomolmed.embopress.org/content/2/9/329>), EMBO Molecular Medicine will publish online a Review Process File (RPF) to accompany accepted manuscripts.

This file will be published in conjunction with your paper and will include the anonymous referee reports, your point-by-point response and all pertinent correspondence relating to the manuscript. Let us know whether you agree with the publication of the RPF.

I look forward to receiving your revised manuscript.

Yours sincerely,

Lise Roth

***** Reviewer's comments *****

Referee #2 (Comments on Novelty/Model System for Author):

The technical quality of the manuscript is extremely good, including CRISPR screens and validation studies done both in vitro and in vivo. The concerns with novelty have been raised with previous rounds of review; however, in my opinion the quality/rigor of this work warrants its publication.

Referee #2 (Remarks for Author):

All of my concerns have been addressed. I recommend accepting this manuscript.

Referee #3 (Remarks for Author):

The revised version makes it easier for me to understand the rationale than the previous one. I was initially confused because I couldn't find the connections between the different sections, but now I realize that there were no direct links - they simply represented different contexts.

Revised version of Ge et al. “Mechanisms of resistance to VHL loss-induced genetic and pharmacological vulnerabilities”**Point-by-point response to referee comments**

We would like to thank the referees for taking the time to review our work. We really appreciate the constructive comments that have allowed us to significantly improve our study.

Referee #2:

“The technical quality of the manuscript is extremely good, including CRISPR screens and validation studies done both in vitro and in vivo. The concerns with novelty have been raised with previous rounds of review; however, in my opinion the quality/rigor of this work warrants its publication.”

Reply: We thank the referee for the positive comments and taking the time to review our work.

Referee #3:

“The revised version makes it easier for me to understand the rationale than the previous one. I was initially confused because I couldn't find the connections between the different sections, but now I realize that there were no direct links - they simply represented different contexts.”

Reply: We thank the referee for taking the time to review our work, and for the constructive comments that have allowed us to substantially improve our study. Indeed, our results show the consequences of VHL inactivation in independent clinical contexts where VHL loss is relevant. We have amended the introduction slightly to clarify this point further.

20th Nov 2025

Dear Dr. Vanharanta,

Thank you for submitting your revised files. I have looked at everything and am ready to accept your manuscript once the following remaining editorial issues are addressed:

- Please define the error bars in the legend of figure EV6G.
- Several figures contain error bars based on $n=2$. Please use scatter blots showing the individual datapoints in these cases and remove the error bars.

Thank you for bearing with these remaining requests.
I look forward to receiving your revised manuscript as soon as possible.

With kind regards,

Lise Roth

The authors addressed the remaining formatting issues.

1st Dec 2025

Dear Dr. Vanharanta,

Thank you for submitting your revised files. I am pleased to inform you that your manuscript is accepted for publication and is now being sent to our publisher to be included in the next available issue of EMBO Molecular Medicine.

If you have any questions, please do not hesitate to contact the Editorial Office.

Thank you for your contribution to EMBO Molecular Medicine!

With kind regards,

Lise Roth
